# HOW TO CRAFT BACKDOORS WITH UNLABELED DATA ALONE?

## ABSTRACT

Relying only on unlabeled data, Self-supervised learning (SSL) can learn rich features in an economical and scalable way. As the drive-horse for building foundation models, SSL has received a lot of attention recently with wide applications, which also raises security concerns where backdoor attack is a major type of threat: if the released dataset is maliciously poisoned, backdoored SSL models can behave badly when triggers are injected to test samples. The goal of this work is to investigate this potential risk. We notice that existing backdoors all require a considerable amount of *labeled* data that may not be available for SSL. To circumvent this limitation, we explore a more restrictive setting called no-label backdoors, where we only have access to the unlabeled data alone, where the key challenge is how to select the proper poison set without using label information. We propose two strategies for poison selection: clustering-based selection using pseudolabels, and contrastive selection derived from the mutual information principle. Experiments on CIFAR-10 and ImageNet-100 show that both no-label backdoors are effective on many SSL methods and outperform random poisoning by a large margin.

## 1 INTRODUCTION

Deep learning's success is largely attributed to large-scale labeled data (like ImageNet (Deng et al., 2009)) which requires a lot of human annotation. However, data collection is often very expensive or requires domain expertise. Recently, Self-Supervised Learning (SSL) that only utilizes unlabeled data has shown promising results, reaching or exceeding supervised learning in various domains (Chen et al., 2020a; Devlin et al., 2019), and serving as foundation models for many downstream applications (Brown et al., 2020). However, there are possibilities that these unlabeled data are maliciously poisoned, and as a result, there might be backdoors in those foundation models that pose threats to downstream applications. Figure 1 outlines such a poisoning scenario, where a malicious attacker injects backdoors to a public unlabeled dataset, *e.g.,* Wiki-text (Merity et al., 2017) and CEM500K (Conrad & Narayan, 2021), and released these poisoned datasets to the users.

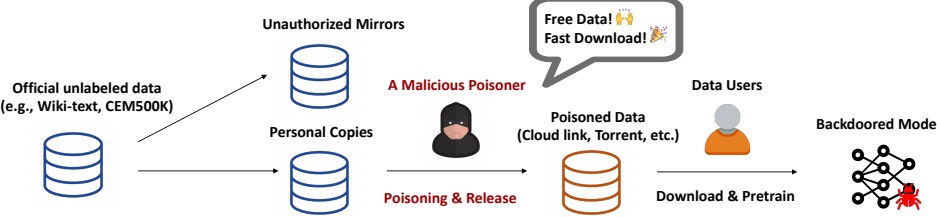

Figure 1: A practical poisoning scenario when the attacker only has access to unlabeled data alone.

To implement such an attack, there remains an important technical problem yet unexplored: **how to inject backdoors with unlabeled data alone?** Up to our knowledge, all existing backdoors require knowing the labels of the poisoned samples (elaborated later). However, in cases like Figure 1, the attacker only has access to unlabeled data alone when poisoning SSL. To circumvent this obstacle, in this paper, we explored a new backdoor scenario called **no-label backdoor (NLB)**. Going beyond dirty-label backdoor (knowing and editing both inputs and labels) (Gu et al., 2017) and clean-label

backdoor (knowing labels, editing only inputs) (Turner et al., 2019), no-label backdoor (knowing and editing only inputs) is a further step towards using minimal information to craft backdoor attacks.

Previous practice of label-aware backdoor reveals that the key to backdoor attack is to induce a strong correlation between the backdoor trigger and a specific target class. Thus, the main obstacle to no-label backdoors is how to inject triggers into a subset of samples with consistent labels. A natural idea is to annotate "pseudolabels" by clustering algorithms like K-means, and use these pseudolabels for backdoor injection. Although effective to some extent, we find that clustering algorithms are often unstable and lead to poor performance under bad initialization. To circumvent this limitation, we develop a new mutual information principle for poison selection, based on which we derive a direct selection strategy called Contrastive Selection. In particular, contrastive selection chooses a subset with maximal similarity between samples within the group (*i.e.,* positive samples), while having minimal similarity to the other samples (*i.e.,* negative samples). Empirically, we find that contrastive selection can produce high class consistency between chosen samples, and as a deterministic method, it is more stable than clustering-based selection. Contributions are summarized below:

- We explore a new scenario of backdoor attack, no-label backdoor (NLB), where the attacker has access to unlabeled data alone. Unlike dirty-label and clean-label backdoors, NLB represents a more restricted setting that no existing backdoors can be directly applied to.
- To craft effective no-label backdoors, we propose two effective strategies to select the poison set from the unlabeled data: clustering-based NLB and contrastive NLB. The former is based on K-means-induced pseudolabels while the latter is directly derived from the proposed mutual information principle for poison selection.
- Experiments on CIFAR-10 and ImageNet-100 show both no-label backdoors are effective for poisoning many SSL methods and outperform random selection significantly by a large margin. Meanwhile, these no-label backdoors are also resistant to finetuning-based backdoor defense to some extent.

**Related Work.** Recently, a few works explore backdoor attack on SSL. Specifically, Saha et al. (2021) propose to poison SSL by injecting triggers to images from the target class, while BadEncoder (Jia et al., 2021) backdoors a SSL model by finetuing the model on triggered target class images. Li et al. (2023) design a color space trigger for backdooring SSL models. However, *all these methods require poisoning many samples from the target classes (e.g., 50% in Saha et al. (2021))*, which is hard to get for the attacker that only has unlabeled data. In this paper, we explore the more restrictive setting when no label information is available for the attacker at all. Besides, Carlini & Terzis (2022) recently studied the backdoor attack on CLIP (Radford et al., 2021), a multi-modal contrastive learning method aligning image-text pairs. However, natural language supervision is also unavailable in many unlabeled dataset, such as, CEM500K. To craft poisons under these circumstances, we explore the most restrictive setting where we have no access to any external supervision, and only the unlabeled data alone. As far as we know, we are the first to explore this no-label backdoor setting.

## 2 THREAT MODEL

We begin with the background of self-supervised learning (SSL) and then introduce the threat model of no-label backdoors for SSL.

### 2.1 BACKGROUND ON SELF-SUPERVISED LEARNING

**Data Collection.** As a certain paradigm of unsupervised learning, SSL requires only unlabeled samples for training (denoted as $\mathcal{D} = \{(\boldsymbol{x}_i)\}$), which allows it to cultivate massive-scale unlabeled datasets. Rather than crawling unlabeled data by oneself, a common practice (also adopted in GPT series) is to utilize a (more or less) curated unlabeled dataset, such as 1) Common Crawl, a massive corpus (around 380TB) containing 3G web pages (Brown et al., 2020); 2) JFT-300M, an internal Google dataset with 300M images with no human annotation (Sun et al., 2017); 3) CEM500K, a 25GB unlabeled dataset of 500k unique 2D cellular electron microscopy images (Conrad & Narayan, 2021). The construction and curvation of these massive-scale datasets is still a huge project. As valuable assets, some of these datasets (like Google's JFT-300M) are kept private and some require certain permissions (like ImageNet). Meanwhile, as these datasets are very large, official downloads

could be very slow. To counter these limits, there are unofficial mirrors and personal links sharing *unauthorized* data copies, which could potentially contain malicious poisons as shown in Figure 1.

**Training and Evaluation.** The general methodology of self-supervised learning is to learn meaningful data representations using self-generated surrogate learning signals, *a.k.a.* self-supervision. Contrastive learning is a state-of-the-art SSL paradigm with well-known examples like SimCLR (Chen et al., 2020a), MoCo (He et al., 2020), and we take it as an example in our work. Specifically, contrastive learning generates a pair of positive samples for each original data $x$ with random data augmentation, and applies the InfoNCE loss Oord et al. (2018):

$$\ell_{nce}(\boldsymbol{z}_i, \boldsymbol{z}_j) = -\log \frac{\exp\left(\text{sim}\left(\boldsymbol{z}_i, \boldsymbol{z}_j\right)/\tau\right)}{\sum_{k=1}^{2B} \mathbf{1}_{[k \neq i]} \exp\left(\text{sim}\left(\boldsymbol{z}_i, \boldsymbol{z}_k\right)/\tau\right)}, \tag{1}$$

where $(\boldsymbol{z}_i, \boldsymbol{z}_j)$ denote the representations of the positive pair that are pulled together, and the others in the minibatch are regarded as negative samples and pushed away. Here, $\tau$ is a temperature parameter and $\text{sim}$ calculates the cosine similarity between two samples. After pretraining, we usually train a linear classifier with labeled data on top of *fixed* representations, and use its classification accuracy (*i.e.,* linear accuracy) to measure the representation quality.

## 2.2 THREAT MODEL OF NO-LABEL BACKDOORS

In this part, we outline the threat model of no-label backdoors (NLB) that tries to inject backdoors to a given unlabeled dataset without label information.

**Attacker's Knowledge.** As illustrated in Figure 1, the attacker can be someone who provides third-party download services of unauthorized copies of a common unlabeled dataset $\mathcal{D}$. This dataset is constructed by others *a priori*, *e.g.,* the medical dataset CEM500K. By holding a clean copy of the unlabeled data $\mathcal{D} = \{x\}$, the attacker has access to all samples in the dataset. However, since he is not the *creator* of the dataset, he does not have the knowledge of the data construction process nor has the domain expertise to create labels for this dataset.

**Attacker's Capacity.** By holding the unlabeled dataset, the attacker can select a small proportion (not exceeding a given budget size $M$) from the unlabeled data as the poison set, and inject stealthy trigger patterns to this subset, keeping other data unchanged. Afterward, he could release the poisoned dataset to the Internet and induce data users to train SSL models with it. The key challenge of crafting no-label backdoors is to **select the proper poison set from the unlabeled dataset**.

**Attack's Goal.** Since the attacker has no label information, he or she cannot craft backdoor attack to a target class as conventional setting. Instead, the attacker can aim at "*untargeted backdoor attack*", where the goal is simply paralyzing the SSL model such that it has poor performance (like untargeted adversarial attack). In this case, the attacker hopes the poisoned model to have high accuracy on clean test data (*i.e.,* high clean accuracy), and low accuracy when injecting triggers to test data (*i.e.,* low poison accuracy). We can also regard the class that most backdoored samples are classified as a "pseudo target class", and calculate attack success rate (ASR) *w.r.t.* this class (higher the better).

## 3 PROPOSED NO-LABEL BACKDOORS

To select the proper poison set without label information, we propose two methods: an indirect method based on clustering-based pseudolabeling (Section 3.1), and a direct method derived from the mutual information principle (Section 3.2).

## 3.1 CLUSTERING-BASED NLB WITH PSEUDOLABELING

**Pseudo-labeling with SSL features.** As label information is absent, a natural way to craft no-label backdoors is to annotate unlabeled data with "pseudolabels" obtained through a clustering algorithm. To obtain better clustering performance, we first utilize a pretrained deep encoder (obtained with a modern SSL algorithm, *e.g.,* SimCLR (Chen et al., 2020a)) to extract feature embeddings for each sample, and then apply the well-known K-means algorithm (Hartigan & Wong, 1979) to clustering these embeddings to $K$ clusters. We treat each cluster as a pseudo-class, and correspondingly, we annotate the unlabeled dataset with $K$ pseudo-labels, $\tilde{\mathcal{D}} = \{(x_i, c_i)\}$, where $c_i \in \mathcal{C} = \{1, \dots, K\}$

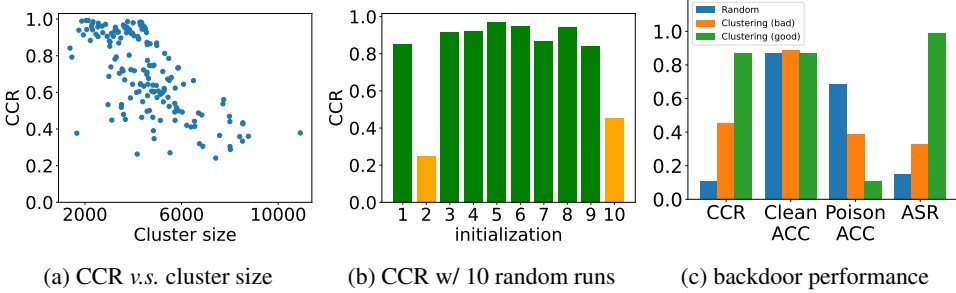

(a) CCR *v.s.* cluster size     (b) CCR w/ 10 random runs     (c) backdoor performance

Figure 2: Analysis on clustering-based NLB on CIFAR-10. a) Cluster Consistency Rate (CCR) of different clusters (obtained with multiple runs). b) CCR across 10 random runs. c) Backdoor performance with three no-label backdoors: random selection, clustering with good CCR (86.93%), clustering with bad CCR (45.48%).

denotes the belonging cluster of $x_i$. In practice, we find that choosing $K$ as the same number of ground-truth labels (*e.g.,* $K = 10$ for CIFAR-10) often yields the best performance.

After pseudo labeling, following the practice of label-aware backdoors, the next step is to select a subset of samples from the same pseudoclass as the poisoning candidates. We measure the quality of each cluster by its class consistency rate (CCR), the ratio of samples from the most frequent class in each cluster. A higher CCR means that the cluster is almost from the same class.

**Poisoning Cluster Selection and Limitations.** Figure 2 summarizes the influence of clustering on CCR. Firstly, we observe that we often get very imbalanced clusters with K-means, ranging from 2,000 to more than 10,000 samples (Figure 2a), and there is a consistent trend, that a larger cluster generally has a smaller CCR. Based on this observation, we select the smallest cluster $\mathcal{D}_c$ whose size exceeds the poison budget $M$ to obtain a more class-consistent poison set. Figure 2b shows that at most time, this strategy yields a much higher CCR than random selection (10% CCR), and also outperforms random selection at backdoor attacks (Figure 2c). Nevertheless, from Figure 2b, we also observe that K-means is unstable and sometimes produce poison sets with very low CCR (about 20%), which is close to random selection. As a result, the clustering approach occasionally produces poor results at bad initialization, as shown in Figure 2c. To resolve this problem, in the next section, we explore a direct approach for poisoning subset selection without using pseudolabels.

## 3.2 CONTRASTIVE NLB WITH MAXIMAL MUTUAL INFORMATION

As discussed in Section 3.1, whether clustering-based NLB succeeds depends crucially on the quality of pseudolabels that varies across different initializations. Thus, we further devise a more principled strategy to directly select the poison set $\mathcal{P} \subseteq \mathcal{D}$ via the maximal mutual information principle.

The mutual information principle is a foundational guideline to self-supervised learning: the learned representations $Z$ should contain the most information of the original inputs; mathematically, the mutual information between the input $X$ and the representation $Z$, *i.e.,*

$$I(X; Z) = \mathbb{E}_{P(X,Z)} \log \frac{P(X, Z)}{P(X)P(Z)}, \tag{2}$$

should be maximized. Many state-of-the-art SSL algorithms, such as contrastive learning and mask autoencoding, can be traced back to the mutual information principle.

**Backdoor as a New Feature.** First, we note that by poisoning a subset of training samples with backdoors and leaving others unchanged, we essentially create a new feature in the dataset. Specifically, let $S$ denote a binary variable that indicates poison selection: given a sample $x \sim X$, we assign

$$P(S|X = x) = \begin{cases} 1, & \text{if } x \in \mathcal{P} \text{ (embedded with backdoor pattern)}; \\ 0, & \text{if } x \notin \mathcal{P} \text{ (no backdoor pattern)}. \end{cases} \tag{3}$$

Thus, any choice of poison set $\mathcal{P}$ will induce a feature mapping between the input trigger and the selection variable $S$, which we call a *backdoor feature*. Although one can simply choose a random subset from $\mathcal{D}_u$, it is often not good enough for crafting effective backdoors (see Figure 2c). This

leads us to rethink the problem about what criterion a good backdoor feature should satisfy, especially without access to label information.

**A Mutual Information Perspective on Backdoor Design.** From the feature perspective, the success of backdoor attacks lies in whether the poisoned data can induce the model to learn the backdoor feature. Thus, to make a strong poison effect, the backdoor features should be designed to be as easy to learn as possible, which can be achieved by encouraging a strong dependence between the backdoor feature and the learning objective, as measured by their mutual information. Since SSL maximizes $I(X; Z)$ during training, denoting $Z^* = \arg\max_Z I(X; Z)$, the backdoor feature should be chosen to maximize their joint mutual information as follows,

$$S^* = \arg\max_S I(X; Z^*; S), \text{ where } I(X; Z^*; S) = I(X; Z^*) - I(X; Z^*|S), \tag{4}$$

and $I(X; Z|S) = \mathbb{E}_{P(X,Z|S)} \log \frac{P(X,Z|S)}{P(X|S)P(Z|S)}$ denotes the conditional mutual information.

*Remark* 3.1. This mutual information principle is general and not limited to no-label backdoors. Notably, it can explain why we need to inject poisons to the same class $Y$ in label-aware backdoors, since $Y$ and $S$ now have high mutual information, which helps achieve a high $I(X; Y; S)$.

To solve Eq. (4) for no-label backdoors, we can first leverage a well-pretrained SSL model and assume that it approximately maximizes the mutual information, *i.e.,* $I(X; Z) \approx I(X, X) = H(X)$, where $H(X) = -\mathbb{E}_{P(X)} \log P(X)$ denotes the Shannon entropy[1]. Then, we can simplify $I(X; Z; S) \approx H(X) - H(X|S) = I(X; S)$. As a result, our ultimate guideline to unlabeled poison selection is to maximize the mutual information between the input and the backdoor, *i.e.,*

$$S^* = \arg\max_S I(X; S), \text{ where } I(X; S) = H(X) - H(X|S). \tag{5}$$

Guided by this principle, the poison selection (indicated by $S$) should create a data split with minimal input uncertainty within each group, *i.e.,* $H(X|S)$. In fact, Eq. (5) is also known as the *Information Gain* criterion used in decision trees when choosing the most important input feature for splitting the data at each node, *e.g.,* in ID3 (Quinlan, 1986). Differently, for backdoor attacks, we have a certain poisoning budget to *create a new backdoor feature $S$* such that the information gain is maximized.

**Tractable Variational Lower Bound.** However, the mutual information $I(X, S) = \mathbb{E}_{P(X,S)} \log \frac{P(X,S)}{P(X)P(S)}$ is computationally intractable since we do not know $P(X)$. To deal with this issue, a common practice is to maximize a tractable variational lower bound of the mutual information (Oord et al., 2018; Hjelm et al., 2019; Poole et al., 2019). Among them, InfoNCE (Oord et al., 2018) is a popular choice and is successfully applied in contrastive learning. Generally speaking, a larger InfoNCE indicates a larger similarity between positive samples (poisoned samples in NLB), and a smaller similarity between negative samples (between poisoned and non-poisoned samples). Following this principle, we design an InfoNCE-style criterion for the poison set $\mathcal{P}$ of budget size $M$,

$$\mathcal{L}_{\text{NCE}}(\mathcal{P}) = \sum_{x \in \mathcal{P}} \sum_{x^+ \in \mathcal{P}} f(x)^\top f(x^+) - \sum_{x \in \mathcal{P}} \log \sum_{x' \notin \mathcal{P}} \exp(f(x)^\top f(x')), \tag{6}$$

where $f(\cdot)$ is a variational encoder (we adopt a pretrained SSL model in practice) and $\tau > 0$ is a temperature hyperparameter. When the dataset is large, the number of negative samples in $X \backslash \mathcal{P}$ could be large. To reduce computation costs, inspired by the fact that $\log \sum_{x' \notin \mathcal{P}} \exp(f(x)^\top f(x')) \geq \max_{x' \notin \mathcal{P}} f(x)^\top f(x')$, we approximate this term with $M$ negative samples with the largest feature similarities to $x$ (*i.e.,* hard negatives), denoted as $\mathcal{N}_x \subseteq \mathcal{X} \backslash \mathcal{P}$. Thus, we arrive at our poison selection criterion called Total Contrastive Similarity (TCS),

$$\mathcal{L}_{\text{TCS}}(\mathcal{P}) = \sum_{x \in \mathcal{P}} \sum_{x^+ \in \mathcal{P}} f(x)^\top f(x^+) - \sum_{x \in \mathcal{P}} \sum_{x' \in \mathcal{N}_x} f(x)^\top f(x'). \tag{7}$$

Both the positive terms and the negative terms are computed with $M \times M$ pairs of samples.

**Efficient Implementation.** Since directly finding a $M$-sized poison set $\mathcal{P} \subseteq \mathcal{X}$ that maximizes Eq. (7) is still NP-hard, we further reduce it to a sample-wise selection problem for computational

---

[1]For the ease of discussion, this is a common assumption adopted in SSL theory (Tsai et al., 2021).

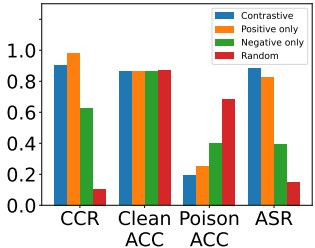 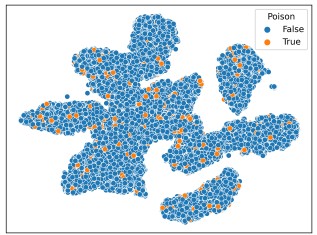 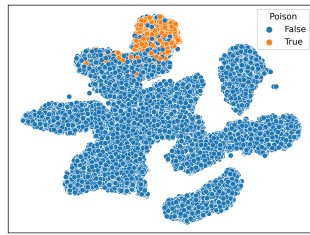

(a) Analysis of contrastive NLB  (b) Random selection samples  (c) Contrastive selection samples

Figure 3: Analysis of contrastive NLB on CIFAR-10. a) backdoor performance of different variants of contrastive selection; b) & c) t-SNE visualization of random and contrastive selection.

efficiency. Specifically, for each sample $x \in \mathcal{X}$, we regard its $M$ nearest samples as the poison set (denoted as $\mathcal{P}_x$), and we choose the poison set $\mathcal{P}_x$ whose anchor sample $x$ has the maximal *sample-wise* contrastive similarity,

$$\ell_{\mathrm{CS}}(x) = \sum_{x^+ \in \mathcal{P}_x} f(x)^\top f(x^+) - \sum_{x' \in \mathcal{N}_x} f(x)^\top f(x'). \tag{8}$$

In this way, following the mutual information principle, samples belonging to the selected poison set have high similarity with each other and small similarity to the other non-poisoned samples.

To summarize, we call the proposed method as a contrastive no-label backdoor, since it relies on both positive and negative samples to select the poison set (Eq. 8). In Figure 3a, we compare the performance of random, positive-only, negative-only, and contrastive selection, where we can see that the contrastive selection yields the lowest backdoor accuracy and the highest ASR (more ablation results in Appendix B.2). Figures 3b & 3c plot the select subset in the feature space. Compared to random selection, poisoned samples with contrastive selection are indeed similar to each other while separating from the rest, which aligns well with the mutual information principle. More algorithm details can be found in Appendix A.

## 4 EXPERIMENTS

In this section, we evaluated the performance of our proposed no-label backdoor attack method on benchmark datasets and compared it with the baseline method. See additional results in Appendix B.

**Training.** We consider two benchmark datasets: a commonly used small dataset CIFAR-10 (Krizhevsky et al., 2009), and a large-scale dataset ImageNet100 adopted by Saha et al. (2021), which is a 1/10 subset of ImageNet-1k (Deng et al., 2009) with 100 classes. For each dataset, we split the total training set into a pre-training set and a downstream task set (9 : 1). For the former, we discard the labels and use the unlabeled data for self-supervised pretaining, use the latter split to learn a linear classifier on top of the learned representation, and evaluate the classification performance on the test data. For SSL pretraining, we include four well-known SSL methods: SimCLR (Chen et al., 2020a), MoCo v2 (Chen et al., 2020b), BYOL (Grill et al., 2020), and Barlow Twins (Zbontar et al., 2021), with default training hyperparameters, all using ResNet-18 as backbones (He et al., 2016). For pretraining, we train 500 epochs on CIFAR-10 and 300 epochs on ImageNet-100. For linear probing, we train 100 epochs on CIFAR-10 and 50 epochs on ImageNet-100.

**Backdoor Poisoning.** We inject poisons to the unlabeled pretraining set with three no-label backdoors: random selection, K-means-based clustering (Section 3.1), and contrastive selection (Section 3.2). We adopt a simple BadNet (Gu et al., 2017) trigger by default, and these methods only differ by the selection of the poison set. By default, the poison rate is 6% for CIFAR-10 and 0.6% for ImageNet-100.

### 4.1 PERFORMANCE ON BENCHMARK DATASETS

We compare the three no-label backdoor methods (random, clustering, contrastive) in Table 1 (CIFAR-10) and Table 2 (ImageNet-100).

Table 1: Performance of different SSL methods under different no-label backdoors on CIFAR-10.

| Pretraining | Backdoor | CCR | Clean ACC(↑) | Poison ACC(↓) | Poison ASR(↑) |
|---|---|---|---|---|---|
| SimCLR | Random (baseline) | 10.70 | **86.90** | 68.54 | 14.93 |
| | Clustering (ours) | **86.93** | **86.97** | **11.02** | **98.80** |
| | Contrastive (ours) | **90.41** | 86.43 | **19.59** | **88.21** |
| MoCo v2 | Random (baseline) | 10.70 | **87.89** | 79.52 | 9.02 |
| | Clustering (ours) | **86.93** | 87.63 | **41.35** | **44.00** |
| | Contrastive (ours) | **90.41** | 87.65 | **41.03** | **62.07** |
| BYOL | Random (baseline) | 10.70 | **90.47** | 85.56 | 6.12 |
| | Clustering (ours) | **86.93** | 89.93 | **47.71** | **47.87** |
| | Contrastive (ours) | **90.41** | 90.21 | **54.94** | **41.35** |
| Barlow Twins | Random (baseline) | 10.70 | 85.38 | 75.33 | 12.22 |
| | Clustering (ours) | **86.93** | **86.10** | **40.26** | **19.66** |
| | Contrastive (ours) | **90.41** | **85.55** | **42.97** | **44.30** |

Table 2: Performance of different SSL methods under different no-label backdoors on ImageNet-100.

| Pretraining | Backdoor | CCR | Clean ACC(↑) | Poison ACC(↓) | Poison ASR(↑) |
|---|---|---|---|---|---|
| SimCLR | Random (baseline) | 1.71 | **61.96** | 58.54 | 1.17 |
| | Clustering (ours) | **94.30** | **62.16** | **34.74** | **40.30** |
| | Contrastive (ours) | **99.72** | 61.48 | **19.34** | **74.46** |
| MoCo v2 | Random (baseline) | 1.71 | 70.74 | 67.90 | 0.91 |
| | Clustering (ours) | **94.30** | **71.24** | **59.38** | **12.26** |
| | Contrastive (ours) | **99.72** | **70.86** | **38.90** | **50.44** |
| BYOL | Random (baseline) | 1.71 | **67.32** | 53.24 | 6.51 |
| | Clustering (ours) | **94.30** | 67.02 | **51.56** | **23.65** |
| | Contrastive (ours) | **99.72** | **67.22** | **22.90** | **72.38** |
| Barlow Twins | Random (baseline) | 1.71 | **70.76** | 67.84 | 0.93 |
| | Clustering (ours) | **94.30** | **71.16** | **56.76** | **19.79** |
| | Contrastive (ours) | **99.72** | 70.10 | **23.18** | **73.85** |

**Comparing with Random Baseline.** First, we notice that the random selection baseline can hardly craft effective no-label backdoors on both datasets: on CIFAR-10, the ASR is around $10\%$ (random guess); on ImageNet-100, the ASR is around $1\%$ in most cases. In comparison, the proposed clustering and contrastive methods can attain nontrivial backdoor performance, degrading the accuracy by at most 75.85% (86.97% → 11.02%), and attaining around 70% ASR on ImageNet-100. Thus, the two selection strategies are both effective NLBs compared to random selection.

**Comparing Clustering and Contrastive Methods.** Among the two proposed backdoors, we notice that the direct contrastive selection outperforms the clustering approach in most cases, and the advantage is more significant on the large-scale ImageNet-100 dataset. For example, clustering-based NLB only attains 19.66%/19.79% ASR for Barlow Twins on CIFAR-10/ImageNet-100, while contrastive NLB attains 44.3%/73.85% ASR instead. This aligns well with our analysis of the limitation of clustering-based poisoning (Section 3.1), and shows that the proposed mutual information principle (Section 3.2) is an effective guideline for designing NLB.

**Comparing to Label-Aware Backdoor (Oracle).** For completeness, we also evaluate label-aware backdoors (LAB) as an oracle setting for no-label backdoors in Table 3. For a fair comparison, we adopt the same poisoning trigger and poisoning rate, and the only difference is that with label information, LAB can select poisoning samples from exactly the same class. Comparing to SimCLR results of no-label backdoors in Tables 2 & 1, we find that their performance is pretty close. Notably, the gap between label-ware backdoor and contrastive NLB is less than 1% in both poison accuracy and ASR when poisoning SimCLR. Therefore, no-label backdoors can achieve comparable poisoning performance even if we only have unlabeled data alone.

Table 3: Performance of label-aware backdoors (oracle) for poisoning SimCLR.

| Dataset | Poison Class | CCR | Clean ACC(↑) | Poison ACC(↓) | Poison ASR(↑) |
|---|---|---|---|---|---|
| CIFAR-10 | Same as Clustering | **100** | **86.56** | **10.35** | **99.60** |
| | Same as Contrastive | **100** | 86.57 | 14.10 | 95.34 |
| ImageNet-100 | Same as Clustering | **100** | **61.80** | 30.38 | 53.53 |
| | Same as Contrastive | **100** | 61.60 | **18.62** | **74.99** |

## 4.2 EMPIRICAL UNDERSTANDINGS

In this part, we further provide some detailed analyses of no-label backdoors.

**Poison Rate.** We plot the performance of clustering and contrastive NLBs under different poison rates in Figure 4. We can see that ranging from 0.02 to 0.10, both NLBs maintain a relatively high level of CCR, and attains ASR higher than 50% in all cases. In particular, contrastive selection maintains a high degree of class consistency with a high poisoning rate and achieves nearly 100% ASR, while clustering NLB degrades in this case. Instead, clustering is more effective under a medium poisoning rate, and achieves better performance with the range of 0.03 to 0.06.

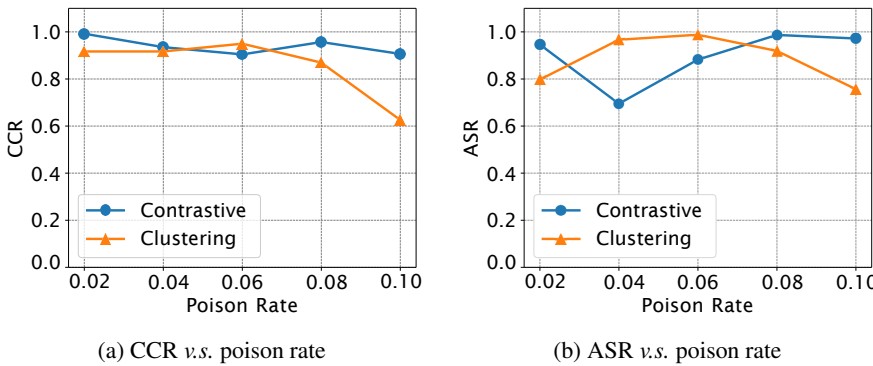

(a) CCR *v.s.* poison rate  (b) ASR *v.s.* poison rate

Figure 4: No-label backdoor on SimCLR with different poisoning rates.

**Trigger Type.** We also evaluate the proposed no-label backdoors with the Blend trigger (Chen et al., 2017), which mixes the poisoned samples with a fixed "Hello Kitty" pattern with a default Blend ratio of 0.2. Table 4 shows that the two proposed methods are also effective with the Blend trigger.

Table 4: Backdoor Performance with the Blend trigger on CIFAR-10.

| Backdoor | Clean ACC (↑) | Poison ACC (↓) | Poison ASR (↑) |
|---|---|---|---|
| Random (baseline) | **86.52** | 47.71 | 26.57 |
| Clustering (ours) | **86.25** | **30.25** | **71.26** |
| Contrastive (ours) | 85.91 | **30.19** | **73.81** |

**Feature-level Interpretation.** At last, we visualize the representations of clean and backdoored data in Figure 5 using a backdoored SimCLR model to understand how the backdoor trigger works in the feature level. We can see that for clean data, the model could learn class-separated clusters (Figure 5a); while for backdoored data, the representations of different classes are mixed together (Figure 5b). It shows that when triggered, the backdoored model (with NLB) can be successfully misled and map all backdoored data to nearly the same place.

**Computation Time.** With a pretrained SSL encoder that is often available online for popular datasets, the poisoning process of no-label backdoors is easy to implement and fast to compute. In practice, with a single GeForce RTX 3090 GPU, clustering and contrastive NLBs can be completed in 9.3 and 10.2 seconds, respectively, showing that we can craft effective no-label backdoors very efficiently.

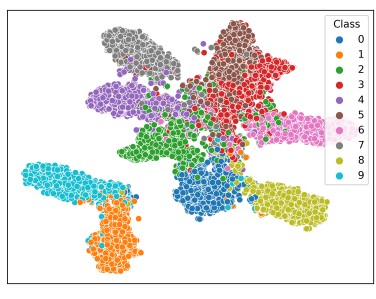 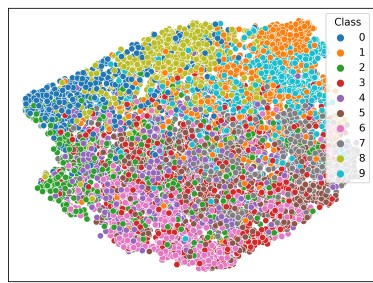

(a) clean samples (without trigger)      (b) backdoored samples (with trigger)

Figure 5: t-SNE visualization on CIFAR-10 test features using a SimCLR encoder attacked by contrastive backdoor. Different colors mark different classes.

## 4.3 Performance under Backdoor Defense

At last, we investigate whether the proposed no-label backdoors can be mitigated by backdoor defense. Finetuning is one of the simplest backdoor defense method, and it is often adopted for transferring pretrained SSL models to downstream tasks. Remarkably, Sha et al. (2022) show that almost all existing backdoors can be successfully removed by a proper configuration of fine-tuning. Here, we explore whether it is also effective for no-label backdoors. We adopt the labeled training split for fully fine-tuning of 30 epochs on CIFAR-10 and 15 epochs on ImageNet-100, respectively.

Comparing the fine-tuning results in Table 5 with the original results in Tables 2 & 1, we find that 1) fine-tuning can indeed defend backdoors by degrading the ASR and improving backdoor accuracy; but 2) meanwhile, the backdoors are not easily removable, since there remains nontrivial ASRs (for example, 83.15% on CIFAR-10 and 24.57% on ImageNet-100) after finetuning, indicating that no-label backdoors are still resistant to finetuning-based defense to some extent.

Table 5: Finetuning results of backdoored SimCLR models on CIFAR-10 and ImageNet-100.

| Dataset | Backdoor | Clean ACC($\uparrow$) | Poison ACC($\downarrow$) | Poison ASR($\uparrow$) |
|---------|----------|------------|-------------|-------------|
| CIFAR-10 | Clustering (ours) | **88.21** | **24.58** | **83.15** |
| | Contrastive (ours) | 88.09 | **35.25** | **70.36** |
| | Label-aware (oracle) | **88.54** | 36.52 | 67.57 |
| ImageNet-100 | Clustering (ours) | **64.14** | 57.98 | 4.18 |
| | Contrastive (ours) | **64.52** | **49.12** | **24.57** |
| | Label-aware (oracle) | **64.14** | 51.56 | 19.54 |

## 5 Conclusion

In this work, we have investigated a new backdoor scenario when the attacker only has access to unlabeled data alone. We explored two new strategies to craft no-label backdoors (NLBs). The first is the clustering-based pseudolabeling method, which can find same-class clusters in most cases, but may occasionally fail due to the unstableness of K-means. To circumvent this limitation, we further propose a new direct selection approach based on the mutual information principle, named contrastive selection. Experiments on CIFAR-10 and ImageNet-100 show that both clustering-based and contrastive no-label backdoors achieve effective backdoor performance, and are resistant to finetuning-based defense to some extent. As further extensions of dirty-label and clean-label backdoors, the proposed no-label backdoors show that backdoor attacks are even possible without using any supervision signals, which poses a meaningful threat to current foundation models relying on self-supervised learning.

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

## A   ALGORITHM DETAILS

In this section, we show the pseudocode of Clustering-based No-label Backdoor and Contrastive No-label Backdoor. For clarity, the code is written in a serial manner, but in actual implementation, both can be easily parallelized using PyTorch on a GPU. Both algorithms have a very short execution time; on CIFAR-10, excluding the pre-training and inference times, they require no more than 3 seconds, significantly shorter than the time needed for pre-training and inference.

Furthermore, if an existing pre-trained encoder is available, the pre-training step can be omitted.

### A.1   PSEUDO CODE OF CLUSTERING-BASED NO-LABEL BACKDOOR

---
**Algorithm 1** Clustering-based No-label Backdoor
---
**input** : data : unlabeled data (N*H*W*C).
           N : size of the unlabeled data.
           M : size of poisoned data.
           $K_c$ : number of clustering categories in the K-means algorithm.
**output** : poison_idx : indices of the poisoned data.

---
1 encoder ← Pretrain(data)
2 feature ← Inference(encoder, data)
3 clusters ← Kmeans(feature, K = $K_c$)
    /* *clusters* are $K_c$ lists, every list has the indices of a clustering
       category                                               */
4 min_cluster_size ← +inf
5 selected_cluster ← none
6 **for** $i \in \{0, 1, \cdots, K_c$-*1*$\}$ **do**
7    | **if** $M \le len(clusters[i]) < min\_cluster\_size$ **then**
8    |   | min_cluster_size ← len (clusters[i])
9    |   | selected_cluster ← clusters[i]

10 poison_idx ← random_sample(selected_cluster, M)
    **return :** poison_idx
---

### A.2   PSEUDO CODE OF CONTRASTIVE NO-LABEL BACKDOOR

---
**Algorithm 2** Contrastive No-label Backdoor
---
**input** : data : unlabeled data (N*H*W*C).
           N : size of the unlabeled data.
           M : size of poisoned data.
**output** : poison_idx : indices of the poisoned data.

---
11 encoder ← Pretrain(data)
12 feature ← Inference(encoder, data)
13 sim_mat ← Matmul(feature, feature.T)
14 **for** $idx \in \{0, 1, \cdots, N$-*1*$\}$ **do**
15    | top2Msim ← TopK_Value(sim_mat[idx,:], K=2M)
16    | score[idx] ← Sum(top2Msim[0:M]) - Sum(top2Msim[M:2M])

17 anchor_idx ← Argmax(score)
18 poison_idx ← TopK_Index(sim_mat[anchor_idx], K=M)
    **return :** poison_idx
---

# B  ADDITIONAL RESULTS

## B.1  PERFORMANCE OF DIFFERENT SSL POISON ENCODER FOR BACKDOOR ATTACKS

In the main text, by default, the attacker adopts the SimCLR-trained model for extracting features used for poison selection. Here, we further compare the performance of different SSL encoders for poison selection. The poisoned data are used for training a SimCLR following the default setting.

From Table 6, we can see that both the four SSL methods considered in this work (SimCLR, MoCo v2, BYOL, Barlow Twins) are comparably effective when used for crafting no-label backdoor.

Table 6: Results of our attack with different SSL threat models and poison models on CIFAR-10.

| Poison | Backdoor | CCR | Clean ACC($\uparrow$) | Poison ACC($\downarrow$) | Poison ASR($\uparrow$) |
|---|---|---|---|---|---|
| SimCLR | Random (baseline) | 10.70 | **86.90** | 68.54 | 14.93 |
| | Clustering (ours) | **86.93** | **86.97** | **11.02** | **98.80** |
| | Contrastive (ours) | **90.41** | 86.43 | **19.59** | **88.21** |
| MoCo v2 | Random (baseline) | 10.70 | **86.90** | 68.54 | 14.93 |
| | Clustering (ours) | **94.63** | 86.85 | **14.74** | **94.44** |
| | Contrastive (ours) | **95.37** | 86.51 | **21.67** | **86.52** |
| BYOL | Random (baseline) | 10.70 | **86.90** | 68.54 | 14.93 |
| | Clustering (ours) | **97.37** | 86.09 | **16.19** | **92.5** |
| | Contrastive (ours) | **88.15** | 86.17 | **12.36** | **97.24** |
| Barlow Twins | Random (baseline) | 10.70 | **86.90** | 68.54 | 14.93 |
| | Clustering (ours) | **83.04** | 86.31 | **21.29** | **84.80** |
| | Contrastive (ours) | **65.44** | 86.33 | **39.15** | **59.16** |

## B.2  COMPARISON OF RANDOM, POSITIVE-ONLY, NEGATIVE-ONLY, AND CONTRASTIVE SELECTION

Here, we present a detailed ablation study of the proposed contrastive selection method in Table 7. We can see that both positive and negative examples contribute to the final backdoor performance. In particular, both terms are necessary on ImageNet-100, since ablating any one of them renders the backdoor ineffective.

Table 7: Results of Random, Positive-only, Negative-only, and Contrastive Selection on CIFAR-10 and ImageNet-100 when poisoning SimCLR.

| Dataset | Backdoor | CCR | Clean ACC($\uparrow$) | Poison ACC($\downarrow$) | Poison ASR($\uparrow$) |
|---|---|---|---|---|---|
| CIFAR-10 | Contrastive | 90.41 | 86.43 | **19.59** | **88.21** |
| | Positive only | **98.33** | 86.71 | 25.04 | 82.48 |
| | Negative only | 62.74 | 86.31 | 40.13 | 39.32 |
| | Random | 10.7 | **86.90** | 68.54 | 14.93 |
| ImageNet-100 | Contrastive | **99.72** | 61.48 | **19.34** | **74.46** |
| | Positive only | 38.75 | **62.06** | 53.84 | 1.79 |
| | Negative only | 44.44 | 61.52 | 44.54 | 3.31 |
| | Random | 1.71 | 61.96 | 58.54 | 1.17 |

## B.3  VISUALIZATION OF DIFFERENT BACKDOORS

We further visualize the features of model trained with different no-label backdoors in Figure 6. For clean data, all models could learn class-separated clusters. For poisoned data, models with no backdoor or random backdoor map them to random classes, while the model with contrastive backdoor can map most poisoned samples to the same class, resulting in consistent misclassification and high ASR.

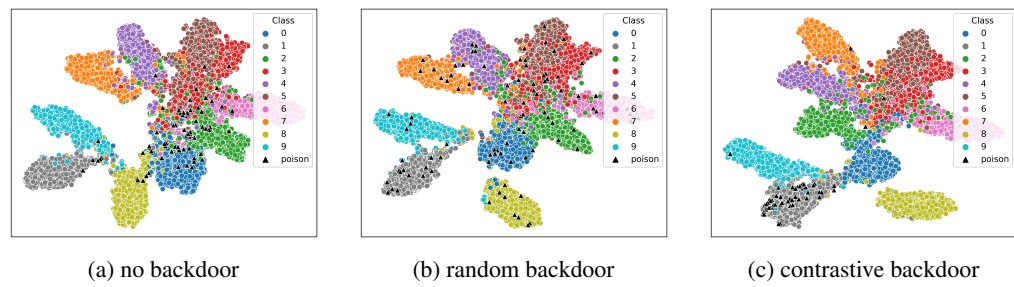

(a) no backdoor          (b) random backdoor         (c) contrastive backdoor

Figure 6: t-SNE feature visualization of all clean test samples and 100 poisoned test samples (with trigger injected) on CIFAR-10. Three models are included: a) a clean model with no backdoor, b) a model with random backdoor, and c) a model with contrastive backdoor.

## B.4    COMPARISON OF MI CRITERIA

As for the mutual information, our TCS objective is derived following our general MI principle. Like InfoNCE an InfoMax, its eventual formulation consists of a positive pair term and a negative pair term, which are the essential ingradients for estimating MI lower bounds. The other simplifications that we used in TCS, such as, truncating the number of negative samples to $M$, are mostly for reducing the overall complexity. The comparison below shows that these different variants of MI-based selection criteria (e.g., including InfoNCE) have similar performance for backdoor poisoning., while TCS has better computation complexity.

Table 8: Comparison of InfoNCE and TCS for contrastive no-label backdoors on CIFAR-10.

| Selection | CCR | Clean Acc | Poison Acc | ASR |
|-----------|------|-----------|------------|-------|
| InfoNCE   | 89.30 | 82.54 | 18.63 | 87.92 |
| TCS       | 90.41 | 86.43 | 19.59 | 88.21 |

## B.5    RESULTS ON MODERN SSL METHODS

To further inverstigate the performance of no-label backdoors for modern SSL methods, we additional evaluate them on DINO (Caron et al., 2021) and MoCo-v3 (Chen et al., 2021) on ImageNet-100.

As shown below in Table 9, our methods also perform well for poisoning DINO and MoCo-v3. In comparison, the contrastive method attains much higher ASR than the clustering method.

Table 9: Evaluation of the two proposed no-label backdoors on ImageNet-100 with DINO and MoCov3.

| Pretraining | Backdoor | CCR | Clean ACC($\uparrow$) | Poison ACC($\downarrow$) | Poison ASR($\uparrow$) |
|-------------|-------------|-------|-----------------------|--------------------------|------------------------|
| DINO        | clustering  | 94.30 | 43.02 | 23.4 | 21.39 |
|             | contrastive | 99.72 | 44.24 | 15.24 | 55.03 |
| MoCov3      | clustering  | 94.30 | 71.5 | 46.9 | 36.78 |
|             | contrastive | 99.72 | 71.34 | 26.46 | 69.79 |

## B.6    TRANSFERABILITY OF BACKDOORS

In the main paper, we evaluate backdoors with on a downstream task with the same label set, which is a common setting among SSL backdoors (Saha et al., 2021). Following your advice, we also evaluate the performance of label-aware and no-label backdoors when transferring from a ImageNet-100 pretrained model to CIFAR-10. As the classes shift, now the task of backdoor attack becomes degrading the poison accuracy as much as possible.

As shown below, label-aware and no-label backdoors are both effective when transferred across datasets. Our contrastive no-label backdoors attain an even lower poison accuracy.

Table 10: Transferability of backdoor attacks across datasets (ImageNet-100 to CIFAR-10).

| Backdoor | Clean ACC($\uparrow$) | Poison ACC($\downarrow$) |
|---|---|---|
| Ours (no-label) | 68.02 | 45.84 |
| Saha et al. (2021) (label-aware) | 69.49 | 47.79 |

## B.7 ADDITIONAL ANALYSIS ON SELECTED POISON SET

To further investigate properties of the poison subset chosen by different methods, we further calculate the label distribution of the poison set (Figure 7a) and the overlap ratio between different poison sets (Figure 7b). We can see that some poison sets have high correlation (e.g., 74%), while some have zero correlation. Upon our further investigation, we find that the high correlation subsets have the same majority class, and vice versa. This provides a useful insight into the poison sets selected by different methods.

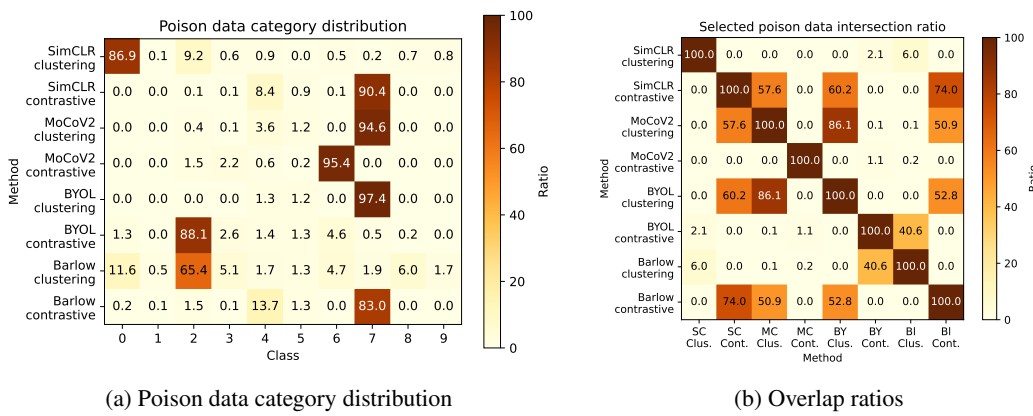

(a) Poison data category distribution      (b) Overlap ratios

Figure 7: Statistics of the poison set selected with different pretraining and poisoning methods.

## B.8 CLUSTERING WITHOUT KNOWLEDGE OF CLUSTER SIZE

We note that the number of ground-truth classes $K$ might be unknown in some unlabeled datasets. Under such circumstances, one can utilize existing strategies to automatically estimate $K$ from data. Here, we try the well-known Silhouette Score (Rousseeuw, 1987). As shown in Table 11 below, $K = 11$ delivers the highest Silhouette Score, which is very close to the ground truth $K = 10$ for CIFAR-10. Table 12 shows that $K = 10$ and $K = 11$ have pretty close CCR, meaning that they are almost equally well for backdoor attack. Therefore, we can indeed get good estimation of $K$ from unlabeled dataset even if it is unknown.

Table 11: Silhouette Score computed with SimCLR representations on CIFAR-10.

| $K$ | 5 | 6 | 7 | 8 | 9 | 10 | 11 | 12 | 13 | 14 |
|---|---|---|---|---|---|---|---|---|---|---|
| Score | 0.066 | 0.071 | 0.0774 | 0.0784 | 0.0814 | 0.0821 | **0.0835** | 0.0825 | 0.0822 | 0.0796 |

Table 12: CCR with different $K$s on CIFAR-10.

| $K$ | 5 | 10 | 11 | 15 | 20 |
|---|---|---|---|---|---|
| CCR | 0.74$\pm$0.22 | 0.82$\pm$0.19 | **0.83$\pm$0.15** | 0.70$\pm$0.24 | 0.70$\pm$0.25 |

### B.9 PERFORMANCE AGAINST PATCHSEARCH DEFENSE

Besides the general finetuning-based defense explored in Section 4.3, we further evaluate no-label backdoors on a defense algorithm specified for SSL, named PatchSearch (Tejankar et al., 2023). We also include label-aware backdoors (Saha et al., 2021) as an oracle baseline.

As shown in Table 13, PatchSearch outperforms finetuning-based defense and is effective for both label-aware and our no-label backdoors. More specifically, our no-label backdoors are more resistant to PatchSearch defense than label-aware methods since both methods attain higher ASRs.

Table 13: Performance of different SSL backdoors against PatchSearch defense on ImageNet-100.

| Method | Clean ACC(↑) | Poison ACC(↓) | Poison ASR(↑) |
|---|---|---|---|
| Label-aware | 61.36 | 55.86 | 1.68 |
| Contrastive NLB | 61.44 | 55.94 | 2.44 |
| Clustering NLB | 61.28 | 55.96 | 2.38 |

## C EXPERIMENT DETAILS

**Backdoor Injection.** We inject BadNets/Blend trigger following the common practice. For CIFAR-10, we add a small 3x3 trigger in the poisoned images for BadNets, and add a "hello kitty" pattern for Blend with a blend ratio of 0.2. For ImageNet-100, we randomly paste a fixed patch to a random position on the image, and make its length and width equal to 1/6 of the image. We show examples of the clean and poisoned images in Figure 8.

**Evaluation Metrics.** We define the evaluation metrics adopted in this work below.

**CCR.** This metric is used to assess whether the selected category has consistent labels. Assuming we have a sample size of $M$, and $C$ is the number of classification categories. Denote a pair of examples as $(x_i, y_i)$, where $x_i$ is the input and $y_i$ is the label. For the $c$-th category, $\text{CCR}_c$ is calculated by:

$$\text{CCR}_c = \sum_{i=0}^{M-1} \mathbf{1}[\hat{y}(x_i) = c]/M.$$

And CCR is calculated by:

$$\text{CCR} = \max_{c \in [0, C-1]} \text{CCR}_c.$$

**ASR.** This metric is used to assess whether the targets of attacks are concentrated. We apply triggers to the evaluation dataset to generate a poison dataset. Let $N$ be the size of the dataset, $C$ be the number of classification categories, and $f(x_i)$ represent the predicted label of the i-th sample. For the c-th category, $\text{ASR}_c$ is calculated by:

$$\text{ASR}_c = \left( \sum_{i=0}^{N-1} \mathbf{1}[(y_i \neq c) \wedge (f(x_i) = c)] \right)/\left( \sum_{i=0}^{N-1} \mathbf{1}[y_i \neq c] \right).$$

And ASR is calculated by:

$$\text{ASR} = \max_{c \in [0, C-1]} \text{ASR}_c.$$

For all experiments in the paper (aside from random poison method), the value of $c$ that maximizes $\text{ASR}_c$ is the same as the value of c that maximizes $\text{CCR}_c$.

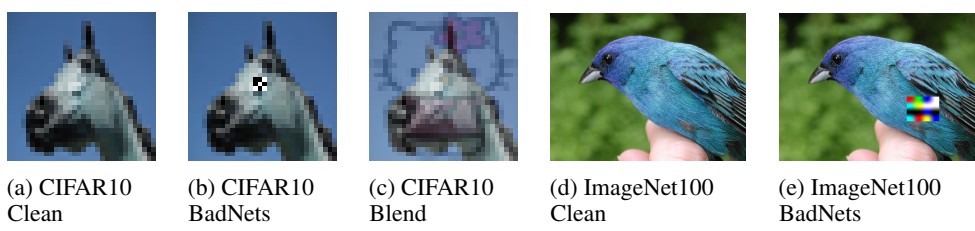

(a) CIFAR10
Clean

(b) CIFAR10
BadNets

(c) CIFAR10
Blend

(d) ImageNet100
Clean

(e) ImageNet100
BadNets

Figure 8: Clean samples and poisoned samples instances in CIFAR-10 and ImageNet-100.

