# OpenReview forum: "How to Craft Backdoors with Unlabeled Data Alone?"
_ICLR.cc/2024/Conference — Submitted to ICLR 2024_

### Official Review · Reviewer_Dh81 · 2023-10-16

**Soundness:** 2 fair
**Presentation:** 3 good
**Contribution:** 2 fair
**Rating:** 3
**Confidence:** 4

**Summary:**

The paper presents an approach for crafting backdoor attacks on datasets without access to label information. Namely, the authors focus on datasets used to train SSL models. In order to achieve this goal, the authors propose several techniques, such as clustering the data points in some latent space and choosing a random cluster, or 2) choosing samples that have high mutual info with each other, and low mutual info with other samples. The authors evaluate their method on 2 datasets: CIFAR10 and ImageNet-100, and implement several SSL methods. They also compare with using labels' information for backdoor attacks, which performed slightly better. Finally the authors evaluate their method against a fine-tuning defense.

**Strengths:**

- the paper presents an attack against SSL methods, and show its effectiveness on 2 datasets
- the attack using no labels has an ASR a bit lower than the ASR when using labels

**Weaknesses:**

- the authors do not evaluate against previous methods proposed to backdoor contrastive learning methods [1,2,3]
- the authors consider only a single type of defenses, and show that the defense is not 100% effective. however, there exist several defenses that approach this problem differently, and might provide a good defense against this attack, such as [4]


[1] Poisoning Web-Scale Training Datasets is Practical, Carlini et al., 2023
[2] Poisoning the Unlabeled Dataset of Semi-Supervised Learning, Carlini et al., 2023
[3] Poisoning and Backdooring Contrastive Learning, Carlini et al., 2023
[4] Incompatibility Clustering as a Defense Against Backdoor Poisoning Attacks, Jin et al., 2023

**Questions:**

- can you please add an evaluation of the other proposed methods?
- can you add more defenses, that seem suitable against your type of attacks, and evaluate the success of your method when those defenses are used?

---

> ### Author Response · Authors · 2023-11-21
> **Response to Reviewer Dh81**
>
> Thanks for your review. We will address your main concerns as follows.
>
> ---
>
> **Q1.** The authors do not evaluate against previous methods proposed to backdoor contrastive learning methods [1,2,3]
>
> **A1**. Thanks for providing these references! Upon our inverstigation, these works cannot work under the no-label backdoor setting considered in our work, since they all require access to the labels of the poisoned data, which are usually not available for unlabeled datasets like CEM-500k. We have added them in the discussion.
>
> **References:**
>
> [1] Poisoning Web-Scale Training Datasets is Practical, Carlini et al., 2023
>
> [2] Poisoning the Unlabeled Dataset of Semi-Supervised Learning, Carlini et al., 2023
>
> [3] Poisoning and Backdooring Contrastive Learning, Carlini et al., 2023
>
> ---
>
> **Q2**. The authors consider only a single type of defenses, and show that the defense is not 100% effective. however, there exist several defenses that approach this problem differently, and might provide a good defense against this attack, such as [4].
>
> > [4] Incompatibility Clustering as a Defense Against Backdoor Poisoning Attacks, Jin et al., 2023
> >
>
> **A2**.  Thanks for your suggestion! Due to the limit of time, we further evaluate our no-label backdoors and previous label-aware SSL backdoors [1] against PatchSearch [2], an backdoor defense method designed specifically for SSL. Results are shown below.
>
> | Method | Clean ACC(↑) | Poison ACC(↓) | Poison ASR(↑) |
> | --- | --- | --- | --- |
> | Label-aware | 61.36 | 55.86 | 1.68 |
> | Contrastive NLB | 61.44 | 55.94 | 2.44 |
> | Clustering NLB | 61.28 | 55.96 | 2.38 |
>
> We can see that PatchSearch outperforms finetuning-based defense and is effective for both label-aware and our no-label backdoors. More specifically, **our no-label backdoors are more resistant to PatchSearch defense than label-aware backdoor [1]** since both methods attain higher ASRs. We have included it in **Appendix B.9**.
>
> **Reference:**
>
> [1] Saha et al. Backdoor attacks on self-supervised learning. CVPR 2022.
>
> [2] Tejankar et al. Defending Against Patch-based Backdoor Attacks on Self-Supervised Learning. CVPR 2023.
>
> ---
>
> **Q3.** Can you please add an evaluation of the other proposed methods?
>
> **A3.** Please see **A1**.
>
> ---
>
> **Q4.** Can you add more defenses, that seem suitable against your type of attacks, and evaluate the success of your method when those defenses are used?
>
> **A4.** Please see **A2**.
>
> ---
>
> Hope the explanations and extended results could ease your concerns. We respectfully suggest that you could re-evaluate our work based on the updated results. We are very happy to address your remaining concerns on our work.

---

> > ### Comment · Reviewer_Dh81 · 2023-11-22
> >
> > Thank you for the clarification.
> >
> > **A1.** I understand that the other contrastive attacks require some label information. However, they still present a valid setup for an attack where adversaries can upload to the Internet images along with captions. A comparison with these approaches would provide more clarity on the benefits of label information.
> >
> >  **A2.** I appreciate running the additional defense. This gives a bit more confidence in the attack, however, an extensive evaluation against different types of defenses is needed, e.g., defenses that are designed for non-patch attacks, or non-outlier based techniques, etc.

---

### Official Review · Reviewer_yPHh · 2023-10-23

**Soundness:** 2 fair
**Presentation:** 3 good
**Contribution:** 2 fair
**Rating:** 5
**Confidence:** 4

**Summary:**

The paper explores the security implications of Self-supervised Learning (SSL), focusing on backdoor attacks that could arise from maliciously poisoned datasets. Given that SSL thrives on unlabeled data to learn features economically, the authors delve into a no-label backdoor setting, a scenario distinct due to the unavailability of labeled data, which is common in existing backdoors. They introduce two strategies for selecting poison samples: a clustering-based selection utilizing pseudolabels, and a contrastive selection based on the mutual information principle. Through experiments on CIFAR-10 and ImageNet-100, they demonstrate the effectiveness of these no-label backdoors across various SSL methods, showing a significant improvement over random poisoning.

**Strengths:**

- The problem investigated in this paper is quite important. Self-supervised learning (SSL), which has become more popular nowadays, is also vulnerable to backdoor attacks.
- This paper is easy to follow.

**Weaknesses:**

- The technical contribution is limited. This paper claims this is an un-target backdoor attack. However, the attack still samples the poisoning examples from a specific class, which is determined by the pseudo-label or the contrastive selection. The only difference with previous works is that the authors select poisoning example candidates by the clustering results, rather than the labels.
- This paper didn't mention some important works in the SSL backdoor attack (e.g., CTRL [1]). In addition, the author didn't compare the proposed work with SOTA SSL backdoor attacks.
- In the threat model, the authors assume that the attack has no domain expertise to create labels for the dataset. However, for the proposed pseudo-labeling method, the authors use $K$, the same as the number of classes, which is unavailable for the attacker according to the threat model. Therefore, the proposed method is not actionable.
- How can CCR be measured without the label information? CCR is the ratio of samples from the most frequent class in each cluster. If the attacker has no label information, how can he know the number of classes of each cluster and the most frequent class? Maybe the authors want to use it to show the statistics of the feature space, but the writing makes the readers think that the attack uses CCR to choose the cluster.
- In Figure 4, for the clustering method, the ASR first increases and then decreases. The authors should give a reason for this observation.
- The defense discussion is not enough. Fine-tuning is not a SOTA defense, especially in SSL.  The author should consider some more advanced defenses (e.g., ASSET [2]).




[1] Li, Changjiang, et al. "Demystifying Self-supervised Trojan Attacks." arXiv e-prints (2022): arXiv-2210.

[2] Pan, Minzhou, et al. "ASSET: Robust Backdoor Data Detection Across a Multiplicity of Deep Learning Paradigms." arXiv preprint arXiv:2302.11408 (2023).

**Questions:**

Please see the weakness.

---

> ### Author Response · Authors · 2023-11-21
> **Response to Reviewer yPHh (1/2)**
>
> Thank you for your detailed reading and for acknowledging the significance and clarity of our work. We address your main concerns below.
>
> ---
>
> **Q1**. The technical contribution is limited. This paper claims this is an un-target backdoor attack. However, the attack still samples the poisoning examples from a specific class, which is determined by the pseudo-label or the contrastive selection. The only difference with previous works is that the authors select poisoning example candidates by the clustering results, rather than the labels.
>
> **A1.** We note that **the clustering method that you criticized, is actually not the ultimate version of our no-label backdoors**. In fact, in Sec 3.1, we have elaborated the limitation of this approach, particularly its clustering instability. To alleviate these issues of clustering methods, we propose a brand new mutual information (MI) principle for backdoor selection, from which we derive the contrastive selection criterion TCS. Notably, **the MI principle of TCS does not involve class priors**. We haver shown that **contrastive method delivers much better results for complex datasets like ImageNet-100 (Table 2).**
>
> Notably, we find that surprisingly, **even without any class prior,** the poison set found by maximizing MI is still highly class-consistent. It suggests that by nature, **the class-related features are the most effective features for backdoor poisoning in image classification datasets**.
>
> Our method is an untargeted attack since it is unknown to which class the target image will be misclassifed to. Since backdoor attack is injecting to training data, it has to induce a strong correlation that are usable for image classification in order to be learned by the model during training. Therefore, an effective backdoor for an image classification dataset must be class-wise and thus behave like a targeted backdoor.
>
> ---
>
> **Q2**. This paper didn't mention some important works in the SSL backdoor attack (e.g., CTRL [1]). In addition, the author didn't compare the proposed work with SOTA SSL backdoor attacks.
>
> **A2**. Like Saha et al, CTRL is also a label-aware SSL backdoor method that requires knowledge of the labels of poison set, and thus it cannot be applied to our no-label backdoor scenario. We have discussed it in the revision.
>
> ---
>
> **Q3.** In the threat model, the authors assume that the attack has no domain expertise to create labels for the dataset. However, for the proposed pseudo-labeling method, the authors use $K$, the same as the number of classes, which is unavailable for the attacker according to the threat model. Therefore, the proposed method is not actionable.
>
> **A3**. Good question! There are two direct solutions to this.
>
> **First**, if $K$ is unknown, you can directly use the contrastive method! It does not need to know $K$ in advance, and has even superior performance for complex datasets like ImageNet-100.
>
> **Second**, if you still want to use the clustering method when $K$ is unknown, there is a bunch of classic methods/metrics for determining $K$ in the clustering literature, e.g., AIC, BIC. Here, we try the well-known Silhouette Score [1] using the [scikit-learn implementation](http://scikit-learn.org/stable/modules/generated/sklearn.metrics.silhouette_score.html). As shown in Table A below, $K=11$ delivers the highest Silhouette Score, which is very close to the ground truth $K=10$ for CIFAR-10. Table B shows that $K=10$ and $K=11$ have pretty close CCR, meaning that they are almost equally well for backdoor attack. Therefore, **we can indeed get good estimation of $K$ from unlabeled dataset even if it is unknown.**
>
> [1] Rousseeuw, Peter J. "Silhouettes: a graphical aid to the interpretation and validation of cluster analysis." *Journal of computational and applied mathematics* 20 (1987): 53-65.
>
> **Table A**. *Silhouette Score computed with SimCLR representations on CIFAR-10*
>
> | K                | 5       | 6       | 7       | 8       | 9       | 10      | 11      | 12      | 13      | 14      |
> |------------------|---------|---------|---------|---------|---------|---------|---------|---------|---------|---------|
> | Silhouette Score | 0.06639 | 0.07112 | 0.07737 | 0.07841 | 0.08136 | 0.08212 | **0.08346** | 0.08248 | 0.08222 | 0.07959 |
>
> **Table B.** CCR with different $K$s.
>
> | K   | 5             | 10            | 11            | 15            | 25           |
> |-----|---------------|---------------|---------------|---------------|--------------|
> | CCR | 0.74$\pm$0.22 | 0.82$\pm$0.19 | **0.83$\pm$0.15** | 0.70$\pm$0.24 | 0.70$\pm$0.2 |

---

> ### Author Response · Authors · 2023-11-21
> **Response to Reviewer yPHh (2/2)**
>
> **Q4**. How can CCR be measured without the label information? CCR is the ratio of samples from the most frequent class in each cluster. If the attacker has no label information, how can he know the number of classes of each cluster and the most frequent class? Maybe the authors want to use it to show the statistics of the feature space, but the writing makes the readers think that the attack uses CCR to choose the cluster.
>
> **A4**. CCR requires label information. But we highlight the fact that **CCR is not required when performing our no-label backdoor methods,** and it is only for analysis purposes. We have revised the writing following your suggestions.
>
> ---
>
> **Q5.** In Figure 4, for the clustering method, the ASR first increases and then decreases. The authors should give a reason for this observation.
>
> **A5**. Because higher poison rates imply a higher cluster size to be found, according to Figure 2, CCR will decrease a lot under large poison rate, and thus degrade the ASR. We have added this explanation to Sec 4.2. In comparison, contrastive method is much stable under larger poison rates.
>
> ---
>
> **Q6.** The defense discussion is not enough. Fine-tuning is not a SOTA defense, especially in SSL. The author should consider some more advanced defenses (e.g., ASSET [2]).
>
> **A6**. Thanks for your suggestion! Due to the limit of time, we further evaluate our no-label backdoors and previous label-aware SSL backdoors [1] against PatchSearch [2], an backdoor defense method designed specifically for SSL. Results are shown below.
>
> | Method | Clean ACC(↑) | Poison ACC(↓) | Poison ASR(↑) |
> | --- | --- | --- | --- |
> | Label-aware | 61.36 | 55.86 | 1.68 |
> | Contrastive NLB | 61.44 | 55.94 | 2.44 |
> | Clustering NLB | 61.28 | 55.96 | 2.38 |
>
> We can see that PatchSearch outperforms finetuning-based defense and is effective for both label-aware and our no-label backdoors. More specifically, **our no-label backdoors are more resistant to PatchSearch defense than label-aware backdoor [1]** since both methods attain higher ASRs. We have included it in **Appendix B.9**.
>
> **Reference:**
>
> [1] Saha et al. Backdoor attacks on self-supervised learning. CVPR 2022.
>
> [2] Tejankar et al. Defending Against Patch-based Backdoor Attacks on Self-Supervised Learning. CVPR 2023.
>
> ---
>
> Thank you again for your review. We have carefully addressed each of your concerns above. We respectfully suggest that you could re-evaluate our work based on these updated results. We are very happy to address your remaining concerns on our work during the discussion stage.

---

### Official Review · Reviewer_w1uC · 2023-11-01

**Soundness:** 3 good
**Presentation:** 2 fair
**Contribution:** 2 fair
**Rating:** 6
**Confidence:** 4

**Summary:**

The paper proposes poison selection strategies for backdoor poisoning in Self-Supervised Learning.

The two proposed strategies are
(1) K-means clustering-based selection using pseudolabels
(2) contrastive selection derived from the mutual information principle

In the first approach, unlabeled data is annotated using “pseudolabels” by clustering algorithms like K-means, and use these cluster pseudolabels for choosing a cluster of samples for backdoor trigger injection.

But as the authors experimental demonstrate, this approach has some limitations. K-means produces imbalanced clusters
and low class consistency rate (CCR) [the ratio of samples from the most frequent class in each cluster].

The second strategy attempts to mitigate the issues above by using a mutual information (MI) formulation. The idea is to associate triggers with chosen samples so that it introduces a good backdoor feature. The MI approach generates poison sets whose members have high similarity with each other and low similarity with other non-poisoned samples.

Experiments demonstrate that the MI approach to select poison samples performs better than random selection and clustering based for poisoning SSL methods like SimCLR, MoCo v2, BYOL and Barlow Twins on datasets CIFAR10 and ImageNet100.

**Strengths:**

The paper attempts to solve the problem of automatically choosing unlabeled samples to add triggers to when poisoning a Self-Supervised Learning (SSL) dataset. Prior work has assumed that information about the poison set is available to the attacker in some form, which ensures similarity of poison set samples. This approach might boost attack performance of Backdoor Attacks against SSL methods.

The paper attempts to solve the problem using a naive approach of k-means clustering and then proposes to use the Mutual Information (MI) formulation to select poison sets. The baselines used are sound, and the experiments are performed on a set of widely used SSL methods.

The writing is good, and the paper is easy to follow.

**Weaknesses:**

The major weaknesses I would like to point out are:

1. The paper is currently missing experiments to show how the selection strategy performs against the defense proposed in this paper.

[A] Tejankar, Ajinkya, et al. “Defending Against Patch-based Backdoor Attacks on Self-Supervised Learning.” Proceedings of the IEEE/CVF Conference on Computer Vision and Pattern Recognition. 2023.

This defense technique is extremely relevant to SSL attacks, and I believe the paper should incorporate this defense in its experiments.

2. The paper does not provide any information about the poison set clusters created by the algorithm. As a starting point, I would like to see a histogram of labels in the poison set chosen and also the target pseudolabel of the poison set. Do the selection strategies pick certain labels? Does that provide us with any information about which classes in the dataset are more likely to be better candidates for adding triggers?

3. (Section 4) The paper says, “We set the default poison rate to 60% of the number of samples per class, which amounts to 6% of CIFAR-10 and 0.6% of ImageNet-100 in total.” But in related work, the paper mentions “However, both methods require poisoning many samples from the target classes (e.g., 50% in Saha et al. (2021)), which is hard to get for the attacker that only has unlabeled data.” It seems the paper ends up using a higher poisoning budget than prior work for their main experiments. Why is such a high poisoning budget required, and does that diminish advantages over prior work?

**Questions:**

Some additional questions which I would like clarification about.

1. (Section 2.2) “calculate attack success rate (ASR) w.r.t. this class (lower the better).” Is lower better or higher better for ASR?

2. (Section 3.1) If it's an unlabeled dataset, how is the K to be chosen? For example, what is K if we want to add No-Label Backdoors to Common Crawl?

3. (Section 3.1) Figure 2(a) seems to show results from multiple K-means runs with K=10. How many times was the K-means clustering repeated?

4. (Section 3.2) To the best of my knowledge, typical multi-view contrastive SSL is based on maximizing mutual information between two augmented views of the same image.

[B] Su, Weijie, et al. “Towards all-in-one pre-training via maximizing multi-modal mutual information.” Proceedings of the IEEE/CVF Conference on Computer Vision and Pattern Recognition. 2023.

[C] Bachman, Philip, R. Devon Hjelm, and William Buchwalter. “Learning representations by maximizing mutual information across views.” Advances in neural information processing systems 32 (2019).

It would be great if there is some clarification regarding whether the above statement is more representative of SSL than “The mutual information principle is a foundational guideline to self-supervised learning: the learned representations Z should contain the most information of the original inputs; mathematically, the mutual information between the input X and the representation Z”.

The authors can provide references to prior work where SSL is formalized as maximization of I(X;Z).

This paper might be relevant.

[D] Tschannen, Michael, et al. “On Mutual Information Maximization for Representation Learning.” International Conference on Learning Representations. 2019.

5. (Section 3.2) The symbol P in Equation 3 seems to be overloaded. P is the poison set, and it also seems to be the binary function. Some clarification regarding the notation will be helpful.

---

> ### Author Response · Authors · 2023-11-21
> **Response to Reviewer w1uC (1/3)**
>
> Thank you for your detailed reading and for acknowledging the novelty and theory of our work. We address your main concerns below.
>
> ---
>
> **Q1**. The paper is currently missing experiments to show how the selection strategy performs against the defense proposed in this paper.
>
> > [A] Tejankar, Ajinkya, et al. “Defending Against Patch-based Backdoor Attacks on Self-Supervised Learning.” Proceedings of the IEEE/CVF Conference on Computer Vision and Pattern Recognition. 2023.
> >
>
> **A1**. Thanks for pointing it out. Following your suggestions, we further evaluate our no-label backdoors and previous label-aware SSL backdoors [B] against PatchSearch [A], an backdoor defense method designed specifically for SSL. Results are shown below.
>
> | Method | Clean ACC(↑) | Poison ACC(↓) | Poison ASR(↑) |
> | --- | --- | --- | --- |
> | Label-aware | 61.36 | 55.86 | 1.68 |
> | Contrastive NLB | 61.44 | 55.94 | 2.44 |
> | Clustering NLB | 61.28 | 55.96 | 2.38 |
>
> We can see that PatchSearch outperforms finetuning-based defense and is effective for both label-aware and our no-label backdoors. More specifically, **our no-label backdoors are more resistant to PatchSearch defense than label-aware backdoor [B]** since both methods attain higher ASRs. We have included it in **Appendix B.9**.
>
> [B] Saha et al. Backdoor attacks on self-supervised learning. CVPR 2022.
>
> ---
>
> **Q2**. The paper does not provide any information about the poison set clusters created by the algorithm. As a starting point, I would like to see a histogram of labels in the poison set chosen and also the target pseudolabel of the poison set. Do the selection strategies pick certain labels? Does that provide us with any information about which classes in the dataset are more likely to be better candidates for adding triggers?
>
> **A2**. Thanks for your suggestion! Following this guide, we calculate the label distribution of the poison set (Table A) and the overlap ratio between different poison sets (Table B). We can see that some poison sets have high correlation (e.g., 74%), while some have zero correlation. Upon our further inverstigation, we find that the high correlation subsets have the same majority class, and vice versa. This provides a useful insight into the poison sets selected by different methods. We have included it to **Appendix B.7**.
>
> **Table A.** Label distribution in selected poison sets.
>
> |  |  | 0 | 1 | 2 | 3 | 4 | 5 | 6 | 7 | 8 | 9 |
> | --- | --- | --- | --- | --- | --- | --- | --- | --- | --- | --- | --- |
> | contras | simclr | 0 | 0 | 0.0011 | 0.0007 | 0.0841 | 0.0093 | 0.0007 | 0.9041 | 0 | 0 |
> | contras | mocov2 | 0 | 0 | 0.0152 | 0.0219 | 0.0063 | 0.0022 | 0.9537 | 0.0004 | 0.0004 | 0 |
> | contras | byol | 0.013 | 0.0004 | 0.8815 | 0.0256 | 0.0137 | 0.0133 | 0.0456 | 0.0048 | 0.0019 | 0.0004 |
> | contras | barlow | 0.0019 | 0.0011 | 0.0152 | 0.0007 | 0.1374 | 0.013 | 0 | 0.8304 | 0.0004 | 0 |
> | cluster | simclr | 0.8693 | 0.0011 | 0.0919 | 0.0063 | 0.0089 | 0 | 0.0052 | 0.0022 | 0.0074 | 0.0078 |
> | cluster | mocov2 | 0.0004 | 0 | 0.0041 | 0.0015 | 0.0356 | 0.0122 | 0 | 0.9463 | 0 | 0 |
> | cluster | byol | 0 | 0 | 0.0004 | 0.0004 | 0.0133 | 0.0122 | 0 | 0.9737 | 0 | 0 |
> | cluster | barlow | 0.1159 | 0.0048 | 0.6544 | 0.0507 | 0.0174 | 0.0133 | 0.0474 | 0.0193 | 0.0596 | 0.017 |
>
> **Table B.** **Confusion matrix of overlap ratios in selected poison sets.
>
> | CIFAR-10 |  | contrastive |  |  |  | cluster |  |  |  |
> | --- | --- | --- | --- | --- | --- | --- | --- | --- | --- |
> |  |  | simclr | mocov2 | byol | barlow | simclr | mocov2 | byol | barlow |
> | contrastive | simclr | 1 | 0 | 0 | 0.7403 | 0.0004 | 0.5762 | 0.6019 | 0 |
> |  | mocov2 | 0 | 1 | 0.0112 | 0 | 0 | 0 | 0 | 0.0022 |
> |  | byol | 0 | 0.0112 | 1 | 0.0004 | 0.021 | 0.0006 | 0.0002 | 0.4059 |
> |  | barlow | 0.7403 | 0 | 0.0004 | 1 | 0 | 0.5088 | 0.5276 | 0.0002 |
> | cluster | simclr | 0.0004 | 0 | 0.021 | 0 | 1 | 0.0004 | 0.0002 | 0.0605 |
> |  | mocov2 | 0.5762 | 0 | 0.0006 | 0.5088 | 0.0004 | 1 | 0.8608 | 0.0006 |
> |  | byol | 0.6019 | 0 | 0.0002 | 0.5276 | 0.0002 | 0.8608 | 1 | 0.0002 |
> |  | barlow | 0 | 0.0022 | 0.4059 | 0.0002 | 0.0605 | 0.0006 | 0.0002 | 1 |

---

> ### Author Response · Authors · 2023-11-21
> **Response to Reviewer w1uC (2/3)**
>
> **Q3.** (Section 4) The paper says, “We set the default poison rate to 60% of the number of samples per class, which amounts to 6% of CIFAR-10 and 0.6% of ImageNet-100 in total.” But in related work, the paper mentions “However, both methods require poisoning many samples from the target classes (e.g., 50% in Saha et al. (2021)), which is hard to get for the attacker that only has unlabeled data.” It seems the paper ends up using a higher poisoning budget than prior work for their main experiments. Why is such a high poisoning budget required, and does that diminish advantages over prior work?
>
> **A3**. We note that the choice of 60% is actually very similar to 50% of Saha, and there is no significant difference between these two. In fact, as we show in Figure 4, our method can attain high ASR by poisoning only 2% of CIFAR-10. So we believe that the poisoning rate is not a drawback of this work.
>
> ---
>
> **Q4**. (Section 2.2) “calculate attack success rate (ASR) w.r.t. this class (lower the better).” Is lower better or higher better for ASR?
> **A4**. Higher the better. Sorry this is a typo. We have fixed it.
>
> ---
>
> **Q5**. (Section 3.1) If it's an unlabeled dataset, how is the K to be chosen? For example, what is K if we want to add No-Label Backdoors to Common Crawl?
> **A5**. Good question! There are two direct solutions to this.
>
> **First**, if $K$ is unknown, you can directly use the contrastive method! It does not need to know $K$ in advance, and has even superior performance for complex datasets like ImageNet-100.
>
> **Second**, if you still want to use the clustering method when $K$ is unknown, there is a bunch of classic methods/metrics for determining $K$ in the clustering literature, e.g., AIC, BIC. Here, we try the well-known Silhouette Score [1] using the [scikit-learn implementation](http://scikit-learn.org/stable/modules/generated/sklearn.metrics.silhouette_score.html). As shown in Table A below, $K=11$ delivers the highest Silhouette Score, which is very close to the ground truth $K=10$ for CIFAR-10. Table B shows that $K=10$ and $K=11$ have pretty close CCR, meaning that they are almost equally well for backdoor attack. Therefore, **we can indeed get good estimation of $K$ from unlabeled dataset even if it is unknown.**
>
> [1] Rousseeuw, Peter J. "Silhouettes: a graphical aid to the interpretation and validation of cluster analysis." *Journal of computational and applied mathematics* 20 (1987): 53-65.
>
> **Table A**. *Silhouette Score computed with SimCLR representations on CIFAR-10*
>
>
> | K                | 5       | 6       | 7       | 8       | 9       | 10      | 11      | 12      | 13      | 14      |
> |------------------|---------|---------|---------|---------|---------|---------|---------|---------|---------|---------|
> | Silhouette Score | 0.06639 | 0.07112 | 0.07737 | 0.07841 | 0.08136 | 0.08212 | **0.08346** | 0.08248 | 0.08222 | 0.07959 |
>
> **Table B.** CCR with different $K$s.
>
> | K   | 5             | 10            | 11            | 15            | 25           |
> |-----|---------------|---------------|---------------|---------------|--------------|
> | CCR | 0.74$\pm$0.22 | 0.82$\pm$0.19 | **0.83$\pm$0.15** | 0.70$\pm$0.24 | 0.70$\pm$0.2 |
>
> ---
>
> **Q6**. (Section 3.1) Figure 2(a) seems to show results from multiple K-means runs with K=10. How many times was the K-means clustering repeated?
>
> **A6**. Here, we repeat 20 times and collect 200 points. We have added this in the appendix.

---

> ### Author Response · Authors · 2023-11-21
> **Response to Reviewer w1uC (3/3)**
>
> **Q7**. (Section 3.2) To the best of my knowledge, typical multi-view contrastive SSL is based on maximizing mutual information between two augmented views of the same image. It would be great if there is some clarification regarding whether the above statement is more representative of SSL than “The mutual information principle is a foundational guideline to self-supervised learning: the learned representations Z should contain the most information of the original inputs; mathematically, the mutual information between the input X and the representation Z”.
>
> The authors can provide references to prior work where SSL is formalized as maximization of I(X;Z). This paper [D] might be relevant.
>
> > [B] Su, Weijie, et al. “Towards all-in-one pre-training via maximizing multi-modal mutual information.” Proceedings of the IEEE/CVF Conference on Computer Vision and Pattern Recognition. 2023.
> >
> >
> > [C] Bachman, Philip, R. Devon Hjelm, and William Buchwalter. “Learning representations by maximizing mutual information across views.” Advances in neural information processing systems 32 (2019).
> >
> > [D] Tschannen, Michael, et al. “On Mutual Information Maximization for Representation Learning.” International Conference on Learning Representations. 2019.
> >
>
> **A7**. Thanks for sharing the literature! In fact, **the founding papers of contrastive learning mostly use the formulation of maximizing $I(X,Z)$ as the learning objective, especially InfoNCE, InfoMax [2], and [3] gives a very good summary on their relationships.** This is because $I(X,Z)$ is a very classic information-theoretical notion for unsupervised representation learning. In the same vein, [D] also considered the $I(X,Z)$ formulation.
>
> We have included these backgrounds in Sec 3.2 in the revision.
>
> **References:**
>
> [1] Oord et al. "Representation learning with contrastive predictive coding." *arXiv preprint arXiv:1807.03748* (2018).
>
> [2] Hjelm et al. "Learning deep representations by mutual information estimation and maximization." *arXiv preprint arXiv:1808.06670* (2018).
>
> [3] Poole et al. "On variational bounds of mutual information." *International Conference on Machine Learning*. PMLR, 2019.
>
> ---
>
> **Q8**. (Section 3.2) The symbol P in Equation 3 seems to be overloaded. P is the poison set, and it also seems to be the binary function. Some clarification regarding the notation will be helpful.
>
> **A8**. In Eq 3, we use $\mathcal{P}$ to denote the probability distribution, and it indeed collides with the poion set. To be consistent with other probability notations in the paper, we have modified it to $P(S|X=x)$.
>
> ---
>
> Thank you again for your careful reading. We hope that the explanations and revisions above could address your concerns. We are very happy to address your remaining concerns during the discussion stage.

---

> > ### Comment · Reviewer_w1uC · 2023-11-22
> > **Response to Rebuttal**
> >
> > I thank the authors for their detailed response to my questions and concerns. It was interesting to see the performance of PatchSearch defense on the attacks. It looks like the defense works quite well.
> >
> > I have no additional questions.

---

### Official Review · Reviewer_gGhQ · 2023-11-03

**Soundness:** 2 fair
**Presentation:** 2 fair
**Contribution:** 2 fair
**Rating:** 3
**Confidence:** 4

**Summary:**

In this work, the authors defined NLB (No-Label Backdoor) attacks, which poisons an unlabeled dataset, such that SSL (Self-Supervised Learning) models trained upon it exhibits both targeted and untargeted backdoor behavior when it is finetuned for a downstream task.
And proposed both a clustering-based and a contrastive method for selecting a subset of training data to perform NLB by injecting BadNet triggers. The proposed methods, especially the contrastive method, significantly outperforms the random baseline across SimCLR, MoCo v2, BYOL and Barlow Twins.

**Strengths:**

1. The NLB attack setting sounds innovative and interesting, given that SSL models are the dominating majority of foundation models.
2. The authors introduced intermediate/intrinsic metrics like CCR (Clustering Consistency Rate) and joint mutual information to assist the comprehension of their poison backdoor selection method.
3. The authors also included experiments to compare NLB with label-aware oracles and to test the resistance to finetuning.

**Weaknesses:**

1. The authors demonstrated extremely limited literature review. They rendered Trojan attacks on SSL models as an underexplored topic while there have been a number of works investigating it [1,2,3,4]. Also, the excuses the authored found to rule out Trojan attacks on Text-Image contrastive learning from comparison isn't convincing. This paper ended up with random selection as the only baseline, giving in a misleading result as if their method performs very well, while a minimum poison rate of 0.2 is by no means acceptable comparing to the commonly used 1% poison rate.
2. The paper lacks a clear demonstration of how their method is being used and evaluated in practice, e.g. how the BadNet/Blend triggers are injected, how ASR is being computed, how CCR, especially for label-aware case, is determined, etc. Their account of mutual information is also very crude and is hardly related to their eventual "efficient" implementation.
3. The experiments are limited to relatively old SSL models and more recent models like DINO, DINO v2, MoCo v3, Mugs, etc are not involved.
4. The experiments are limited to using the same relatively small dataset for both (poison/pre) training and (downstream/defensive) finetuning.
​


References:
​

[1] Changjiang Li, , Ren Pang, Zhaohan Xi, Tianyu Du, Shouling Ji, Yuan Yao, Ting Wang. "An Embarrassingly Simple Backdoor Attack on Self-supervised Learning." (2022).

[2] Jia, Jinyuan, et al. “BadEncoder: Backdoor Attacks to Pre-Trained Encoders in Self-Supervised Learning.” 2022 IEEE Symposium on Security and Privacy (SP), May 2022. Crossref, https://doi.org/10.1109/sp46214.2022.9833644.

[3] Liu, Hongbin, Jinyuan Jia, and Neil Zhenqiang Gong. "{PoisonedEncoder}: Poisoning the Unlabeled Pre-training Data in Contrastive Learning." *31st USENIX Security Symposium (USENIX Security 22)*. 2022.

[4] Saha, Aniruddha, et al. “Backdoor Attacks on Self-Supervised Learning.” 2022 IEEE/CVF Conference on Computer Vision and Pattern Recognition (CVPR), June 2022. Crossref, https://doi.org/10.1109/cvpr52688.2022.01298.
​
​

**Questions:**

1. Why do the authors not compare their attack with any of the existing Trojan attacks against SSL, like [1,2,3,4]. Instead of crafting a backdoor attack, this work sounds more like proposing a clever way of selecting a subset to poison with random and dated triggers targeting old models and small datasets, but the eventual poison rate is remarkably high.
2. How is CCR computed for label-aware methods? Did you mean in Table 3 that you randomly poisoned $M$ samples from the class (in terms of label) of data that is most heavily chosen by the proposed clustering-based/contrastive method?
3. Why label-aware methods only poisoned the class chosen by the proposed method instead of all possible ones given that you are testing the poison subset selection performance and the real oracle choice could fall in a different class?
4. Why would a poison attack on a high CCR subset be expected to generalize to other data? With high CCR, couldn't $H(Z^*|S)$ be very low and causes the poison to be effective for the class that is most heavily impacted only? In Figure 5, many classes are still forming an identifiable cluster despite being drawn closer. Which classes of them contain samples that are being selected by your methods?
5. The same dataset is being used for finetuning and training which doesn't resembles the practical use the most. How about taking a SSL trained model on ImageNet-100 and finetune it on CIFAR-10 and vise versa? What about some other defensive methods like backdoor removal methods.
6. If the choice of SSL model for feature extraction is not relevant as in Table 6, do they result in the same/very similar choice of poison dataset? It will be nice to see their ratio of overlapping part. If they are dissimilar, why? In addition, given that there is a "pseudo target class" eventually (thus high ASR), have you considered using a target class's average feature as an anchor and optimize for a trigger instead of using a fixed one?
7. In Section 2.2 why would lower ASR be the attack's goal? Is that a typo?
​

---

> ### Author Response · Authors · 2023-11-21
> **Response to Reviewer gGhQ (1/3)**
>
> Thank you for your review, but we are afraid that there might be some misunderstanding of the problem setup. We have explained it in detail, and revised our paper following your suggestions.
>
> ---
>
> **Q1.** The authors demonstrated extremely limited literature review.
>
> > They rendered Trojan attacks on SSL models as an underexplored topic while there have been a number of works investigating it [1,2,3,4].
> >
>
> **A1**. We are afraid that you might misunderstand our points here. We do acknowledge the literature of SSL backdoors and **both [2,4] has been discussed in our related work** (Sec 1), and [1] is very similar to [4] since it also requires target-class samples (now included in the discussion). We note that [3] is poisoning attack, not backdoor attack.
>
> Notably, in the introduction, we are NOT claiming “Trojan attacks on SSL models is underexplored”, instead, we claim **“no existing work (including those SSL backdoors) explores Trojan attack with *unlabeled data alone*”,** since they also exploit label/text supervision during poisoning. **Up to our knowledge, the no-label backdoor (NLB) setting of SSL is firstly studied in our work** (please correct us if wrong). This setting is reasonable, because **many real-world unlabeled data, such as, CEM500K, do not contain any target information** for backdoor injection (as assumed in [1,2,4]) and the attackers may not have the domain expertise to select/annotate the images.
>
> > Also, the excuses the authored found to rule out Trojan attacks on Text-Image contrastive learning from comparison isn't convincing.
> >
>
> This is because like labels, natural text is also a form of external supervision (as indicated in CLIP’s title [5]). Thus CLIP is not an SSL method. **Since label / natural text is unavailable in many unlabeled dataset like CEM500K, Calini’s method cannot be applied.** As our title indicates, the goal of this paper is to explore techniques for crafting backdoors **in the no-label scenario that none previous work applies.** We are not trying to outperform other SSL methods in all existing SSL / CL scenarios they have already explored.
>
> > This paper ended up with random selection as the only baseline, giving in a misleading result as if their method performs very well, while a minimum poison rate of 0.2 is by no means acceptable comparing to the commonly used 1% poison rate.
> >
>
> We are afraid this is also an misunderstanding. In the experiment section, we wrote ***“we set the default poison rate to 60% of the number of samples per class, which amounts to 6% of CIFAR-10 and 0.6% of ImageNet-100 in total.***” Accordingly, **the minimum poison rate we studied is 2%, instead of 20%**, when calculating over the entire CIFAR-10 dataset**.** Also note that the default poison rate in Saha et al. [4] is 5%. Figure 4 shows that our method performs quite well under 2% poison rate without target label information.
>
> We acknowledge that our calculation could be a bit confusing. We now change the descriptions and captions to be clearer in the revision.
>
> **References:**
>
> [1] Changjiang Li, , Ren Pang, Zhaohan Xi, Tianyu Du, Shouling Ji, Yuan Yao, Ting Wang. "An Embarrassingly Simple Backdoor Attack on Self-supervised Learning." (2022).
>
> [2] Jia, Jinyuan, et al. “BadEncoder: Backdoor Attacks to Pre-Trained Encoders in Self-Supervised Learning.” 2022 IEEE Symposium on Security and Privacy (SP), May 2022.
>
> [3] Liu, Hongbin, Jinyuan Jia, and Neil Zhenqiang Gong. "{PoisonedEncoder}: Poisoning the Unlabeled Pre-training Data in Contrastive Learning." *31st USENIX Security Symposium (USENIX Security 22)*. 2022.
>
> [4] Saha, Aniruddha, et al. “Backdoor Attacks on Self-Supervised Learning.” 2022 IEEE/CVF Conference on Computer Vision and Pattern Recognition (CVPR), June 2022.
>
> [5] Radford et al. Learning Transferable Visual Models From Natural Language Supervision. In ICML. 2021.
>
> ---
>
> **Q2**. The paper lacks a clear demonstration of how their method is being used and evaluated in practice, e.g. how the BadNet/Blend triggers are injected, how ASR is being computed, how CCR, especially for label-aware case, is determined, etc. Their account of mutual information is also very crude and is hardly related to their eventual "efficient" implementation.
>
> **A2**. Thanks for your suggestions. We now added **Appendix C** to describe the **experiment details, where we give sufficient details.** Specifically, We inject BadNet/Blend trigger following the common practice. We add a small 3x3 trigger in the poisoned images for BadNets, and add a “hello kitty” pattern for Blend with a blend ratio of 0.05. As described in Sec 2.2, the ASR is calculated as the proportion of samples from other classes that are misclassified to “pseudo target class”. In the label-aware setting, we can directly inject all backdoors to a chosen class, so CCR is 100%. Please refer to the paper for more details.
>
> (continue below)

---

> ### Author Response · Authors · 2023-11-21
> **Response to Reviewer gGhQ (2/3)**
>
> As for the **mutual information,** our TCS objective is **derived following our general MI principle**. Like InfoNCE an InfoMax, its eventual formulation consists of a positive pair term and a negative pair term, which are the essential ingradients for estimating MI lower bounds. The other simplifications that we used in TCS, such as, truncating the number of negative samples to $M$, are mostly for reducing the overall complexity. The comparison below shows that **these different variants of MI-based selection criteria (e.g., including InfoNCE) have similar performance for backdoor poisoning., while TCS has better computation complexity.**
>
> | Selection  | CCR | Clean Acc | Poison Acc | ASR |
> | --- | --- | --- | --- | --- |
> | InfoNCE | 89.30 | 82.54 | 18.63 | 87.92 |
> | TCS | 90.41 | 86.43 | 19.59 | 88.21 |
>
> ---
>
> **Q3**. The experiments are limited to relatively old SSL models and more recent models like DINO, DINO v2, MoCo v3, Mugs, etc are not involved.
>
> **A3**. Since previous SSL backdoors (such as Saha’s) also consider the well-known methods, eg SimCLR, MoCo, for benchmarking the backdoor performance. We also follow this convention. As shown below, **our methods also perform well for poisoning DINO and MoCo-v3**. In comparison, the contrastive method attains much higher ASR than the clustering method.
>
> *Evaluation of the two proposed no-label backdoors on ImageNet-100.*
>
> | Pretraining | Backdoor | CCR | Clean ACC(↑) | Poison ACC(↓) | Poison ASR(↑) |
> | --- | --- | --- | --- | --- | --- |
> | DINO | clustering | 94.30 | 43.02 | 23.4 | 21.39 |
> |  | contrastive | 99.72 | 44.24 | 15.24 | 55.03 |
> | MoCov3 | clustering | 94.30 | 71.5 | 46.9 | 36.78 |
> |  | contrastive | 99.72 | 71.34 | 26.46 | 69.79 |
>
> ---
>
> **Q4**. The experiments are limited to using the same relatively small dataset for both (poison/pre) training and (downstream/defensive) finetuning.
>
> **A4**. We note that ImageNet-100 is usually considered as a large scale dataset for backdoor study, since it contains 1M high-resolution images (15G file size). In Saha et al, ImageNet-100 is used as the *largest* scale experiment, and up to our knowledge, no existing SSL backdoors have full ImageNet experiments. Overall, we believe that the ImageNet-100 experiments sufficiently demonstrated the scalability of our methods.
>
> ---
>
> **Q5**. Why do the authors not compare their attack with any of the existing Trojan attacks against SSL, like [1,2,3,4]. Instead of crafting a backdoor attack, this work sounds more like proposing a clever way of selecting a subset to poison with random and dated triggers targeting old models and small datasets, but the eventual poison rate is remarkably high.
>
> **A5**. Please see **A1**.
>
> ---
>
> **Q6**. How is CCR computed for label-aware methods? Did you mean in Table 3 that you randomly poisoned $M$ samples from the class (in terms of label) of data that is most heavily chosen by the proposed clustering-based/contrastive method?
>
> **A6**. As described in Sec 2.2, the ASR is calculated as the proportion of samples from other classes that are misclassified to “pseudo target class”. In the label-aware setting, we can directly inject all backdoors to a chosen class, so **CCR is 100%**.
>
> **Reason**. We note that the label-aware setting serves an oracle baseline in our method. In order to eliminate the performance difference caused by class selection, we choose the class picked by the clustering-based/contrastive method **for a fair comparison of no-label and label-aware backdoors.**
>
> > Why label-aware methods only poisoned the class chosen by the proposed method instead of all possible ones given that you are testing the poison subset selection performance and the real oracle choice could fall in a different class?
> >
>
> **Fixed class selection**. Following your suggestion, we also study the usual setting, that is to pick a fixed class for label-aware poisoning. Due to the limit of time, we consider three classes (0,1,2) in CIFAR-10, and report their results below. We can see that the overall results are similar to Table 3.
>
> |  | Clean ACC(↑) | Poison ACC(↓) | Poison ASR(↑) |
> | --- | --- | --- | --- |
> | 0 | 86.56 | 86.56 | 99.6 |
> | 1 | 83.02 | 10.97 | 98.56 |
> | 2 | 83.23 | 10.06 | 99.93 |

---

> ### Author Response · Authors · 2023-11-21
> **Response to Reviewer gGhQ (3/3)**
>
> **Q8**. Why would a poison attack on a high CCR subset be expected to generalize to other data?  With high CCR, couldn't $H(Z^*|S)$ be very low and causes the poison to be effective for the class that is most heavily impacted only?
>
> **A8**. Intuitively, a high CCR means that the poisoned data almost belong to the same class (similar to label-aware backdoors), which creates a strong correlation between the trigger pattern and the majority class, **making the trigger a useful and easy-to-learn feature for predicting the majority class.** As a result, the trigger pattern will be utilized by the model and thus be embedded into the model. For a poisoned model, **the attacker can misclassify any image to the majority class by simply adding a trigger to it.** Thus, it is effective for samples from all classes. Standard label-aware backdoors can be seen as an oracle no-label backdoor with 100% CCR, and also fit our discussion.
>
> ---
>
> **Q9**. The same dataset is being used for finetuning and training which doesn't resembles the practical use the most. How about taking a SSL trained model on ImageNet-100 and finetune it on CIFAR-10 and vise versa? What about some other defensive methods like backdoor removal methods.
>
> **A9**. We note that the evaluation on a downstream task with the same label set is **a common setting among SSL backdoors, such as Saha et a**l. Following your advice, we also evaluate the performance of label-aware and no-label backdoors when transferring from a ImageNet-100 pretrained model to CIFAR-10. As the classes shift, now the task of backdoor attack becomes degrading the poison accuracy as much as possible.
>
> As shown below, label-aware and no-label backdoors are both effective when transferred across datasets. Our contrastive no-label backdoors attain an even lower poison accuracy.
>
> *Transferability of backdoor attack across datasets (ImageNet-100 to CIFAR-10).*
>
> | Backdoor | Clean ACC(↑) | Poison ACC(↓) |
> | --- | --- | --- |
> | Ours (no-label) | 68.02 | 45.84 |
> | Saha’s (label-aware) | 69.49 | 47.79 |
>
> ---
>
> **Q10.** If the choice of SSL model for feature extraction is not relevant as in Table 6, do they result in the same/very similar choice of poison dataset? It will be nice to see their ratio of overlapping part. If they are dissimilar, why? In addition, given that there is a "pseudo target class" eventually (thus high ASR), have you considered using a target class's average feature as an anchor and optimize for a trigger instead of using a fixed one?
>
> **A10**. That’s a very interesting question! We plot a confusion matrix below that calculates the overlap ratio between the selected poison sets of different methods. We can see that some poison sets have high correlation (e.g., 74%), while some have zero correlation. Upon our further inverstigation, we find that the high correlation subsets have the same majority class, and vice versa. This provides a useful insight into the poison sets selected by different methods. We have included it to **Appendix B.7.**
>
> **Table A.** **Confusion matrix of overlap ratios in selected poison sets.
>
> | CIFAR-10 |  | contrastive |  |  |  | cluster |  |  |  |
> | --- | --- | --- | --- | --- | --- | --- | --- | --- | --- |
> |  |  | simclr | mocov2 | byol | barlow | simclr | mocov2 | byol | barlow |
> | contrastive | simclr | 1 | 0 | 0 | 0.7403 | 0.0004 | 0.5762 | 0.6019 | 0 |
> |  | mocov2 | 0 | 1 | 0.0112 | 0 | 0 | 0 | 0 | 0.0022 |
> |  | byol | 0 | 0.0112 | 1 | 0.0004 | 0.021 | 0.0006 | 0.0002 | 0.4059 |
> |  | barlow | 0.7403 | 0 | 0.0004 | 1 | 0 | 0.5088 | 0.5276 | 0.0002 |
> | cluster | simclr | 0.0004 | 0 | 0.021 | 0 | 1 | 0.0004 | 0.0002 | 0.0605 |
> |  | mocov2 | 0.5762 | 0 | 0.0006 | 0.5088 | 0.0004 | 1 | 0.8608 | 0.0006 |
> |  | byol | 0.6019 | 0 | 0.0002 | 0.5276 | 0.0002 | 0.8608 | 1 | 0.0002 |
> |  | barlow | 0 | 0.0022 | 0.4059 | 0.0002 | 0.0605 | 0.0006 | 0.0002 | 1 |
>
> **Table B.** Label distribution in selected poison sets.
>
> |  |  | 0 | 1 | 2 | 3 | 4 | 5 | 6 | 7 | 8 | 9 |
> | --- | --- | --- | --- | --- | --- | --- | --- | --- | --- | --- | --- |
> | contras | simclr | 0 | 0 | 0.0011 | 0.0007 | 0.0841 | 0.0093 | 0.0007 | 0.9041 | 0 | 0 |
> | contras | mocov2 | 0 | 0 | 0.0152 | 0.0219 | 0.0063 | 0.0022 | 0.9537 | 0.0004 | 0.0004 | 0 |
> | contras | byol | 0.013 | 0.0004 | 0.8815 | 0.0256 | 0.0137 | 0.0133 | 0.0456 | 0.0048 | 0.0019 | 0.0004 |
> | contras | barlow | 0.0019 | 0.0011 | 0.0152 | 0.0007 | 0.1374 | 0.013 | 0 | 0.8304 | 0.0004 | 0 |
> | cluster | simclr | 0.8693 | 0.0011 | 0.0919 | 0.0063 | 0.0089 | 0 | 0.0052 | 0.0022 | 0.0074 | 0.0078 |
> | cluster | mocov2 | 0.0004 | 0 | 0.0041 | 0.0015 | 0.0356 | 0.0122 | 0 | 0.9463 | 0 | 0 |
> | cluster | byol | 0 | 0 | 0.0004 | 0.0004 | 0.0133 | 0.0122 | 0 | 0.9737 | 0 | 0 |
> | cluster | barlow | 0.1159 | 0.0048 | 0.6544 | 0.0507 | 0.0174 | 0.0133 | 0.0474 | 0.0193 | 0.0596 | 0.017 |
>
> ---
>
> **Q11**. In Section 2.2 why would lower ASR be the attack's goal? Is that a typo?
>
> **A11**. Sorry this is a typo. We have fixed it.

---

> ### Author Response · Authors · 2023-11-21
>
> Thank you again for your careful reading. We have carefully refined our paper according to your suggestions, and address each of your concerns above. We respectfully suggest that you could re-evaluate our work based on these updated results. We are very happy to address your remaining concerns on our work.

---

### Author Response · Authors · 2023-11-21
**Paper Update**

We sincerely thank all reviewers for their detailed reading and valuable comments. We have carefully responded their concerns, and incorporated these suggestions in the updated manuscript with 3 extended pages. The main revisions are:

- **Sec 1: add more related works on SSL backdoors**
- Sec 4: fix confusions on poison rate
- **Appendix B.4 (new):** add comparison of MI criteria, InfoNCE and TCS (ours)
- **Appendix B.5 (new):** add two modern SSL methods, DINO and MoCov3
- **Appendix B.6 (new):** add transferability experiments from ImageNet-100 to CIFAR-10
- **Appendix B.7 (new)**: more statistics and analyses on the selected poison set
- **Appendix B.8 (new)**: performance of clustering-based backdoor without knowing $K$
- **Appendix B.9 (new)**: performance against an SSL backdoor defense, PatchSearch
- **Appendix C (new)**: experiment details and trigger visualization

---

### Author Response · Authors · 2023-11-22
**Please leave your comment**

Dear All Reviewers,

We have prepared a detailed response to each of your concerns. Could you please take a moment to look at it, and tell us whether you still have additional concerns?

ICLR has been an active community well-known for its diverse and lively discussions. We are eagerly awaiting your response.

Thanks for your time! Have a great day!

Authors

---

### Meta-Review · Area_Chair_znXa · 2023-12-12

**Metareview:**

(a) Summarize the scientific claims and findings of the paper based on your own reading and characterizations from the reviewers.

This paper explores a restrictive setting called no-label backdoors, where the attacker only has access to the unlabeled data alone, (and so does the model trainer). Two strategies are proposed for poison selection: clustering-based selection using pseudolabels, and contrastive selection derived from the mutual information principle. In the first approach, unlabeled data is annotated using “pseudolabels” by clustering algorithms like K-means, and use these cluster pseudolabels for choosing a cluster of samples for backdoor trigger injection. The second strategy attempts to mitigate the issues above by using a mutual information (MI) formulation. The idea is to associate triggers with chosen samples so that it introduces a good backdoor feature. The MI approach generates poison sets whose members have high similarity with each other and low similarity with other non-poisoned samples. Experiments on CIFAR-10 and ImageNet-100 show that both no-label backdoors are effective on many SSL methods and outperform random poisoning by a large margin.

(b) What are the strengths of the paper?

The new problem of attacking SSL training with no label backdoor attack is novel and interesting. Clustering consistency and mutual information are novel and important contributions.

(c) What are the weaknesses of the paper? What might be missing in the submission?

Important references are missing. Details on the setting, algorithm, and evaluation are missing. The interpretation of the results from clustering is missing. It is misleading to call this attack backdoor attack, since typical backdoor attacks have target label and triggers. There is no target label for this problem, so it is closer to triggerless data poisoning attack, just that there is actually trigger. So in some sense, the problem is different from both triggerless data poisoning attack and backdoor attacks. This should be made clear in the beginning of the paper. More fundamentally, this begs the question of what is the paper trying to achieve? It is not clear what the attacker's goal is at inference time. For example, if the attacker is the one who is adding the trigger at the inference time, why would anyone care about the Poison ACC (other than the adversary maybe). Also, if there is no target label, why would anyone care about ASR?

**Justification For Why Not Higher Score:**

At a first glance, the paper proposes a new problem and provides an interesting solution based on mutual information. But after understanding the evaluations of their experiments further, one can realize that the problem is flawed. The attack studied in this paper is not a real attack in the sense that it is not clear what the risk or harm is in the scenario studied.

**Justification For Why Not Lower Score:**

N/A

---

### Decision · Program_Chairs · 2024-01-16

Reject